# Modeling the Human Visual System: Comparative Insights from Response-Optimized and Task-Optimized Vision Models, Language Models, and different Readout Mechanisms

## Abstract

Over the past decade, predictive modeling of neural responses in the primate visual system has advanced significantly, largely driven by various deep neural network approaches. These include models optimized directly for visual recognition, cross-modal alignment through contrastive objectives, neural response prediction from scratch, and large language model embeddings. Likewise, different readout mechanisms—ranging from fully linear to spatial-feature factorized methods—have been explored for mapping network activations to neural responses. Despite the diversity of these approaches, it remains unclear which method performs best across different visual regions. In this study, we systematically compare these approaches for modeling the human visual system and investigate alternative strategies to improve response predictions. Our findings reveal that for early to mid-level visual areas, response-optimized models with visual inputs offer superior prediction accuracy, while for higher visual regions, embeddings from Large Language Models (LLMs) based on detailed contextual descriptions of images and task-optimized models pretrained on large vision datasets provide the best fit. Through comparative analysis of these modeling approaches, we identified three distinct regions in the visual cortex: one sensitive primarily to perceptual features of the input that are not captured by linguistic descriptions, another attuned to fine-grained visual details representing semantic information, and a third responsive to abstract, global meanings aligned with linguistic content. We also highlight the critical role of readout mechanisms, proposing a novel scheme that modulates receptive fields and feature maps based on semantic content, resulting in an accuracy boost of 3-23% over existing SOTAs for all models and brain regions. Together, these findings offer key insights into building more precise models of the visual system.

## 1 Introduction and Related Work

The effort to build accurate predictive models of the visual system has been a longstanding goal in neuroscience. Early approaches primarily relied on handcrafted features, such as Gabor filters, curvature models, and motion energy models, to predict responses in early to mid-level visual areas. Similarly, word-based descriptions were often used to model responses in higher-level visual regions. These models provided interpretability, as the features they employed were well understood and linked to specific visual computations. However, they lacked quantitative precision in their ability to predict neural responses (Hubel and Wiesel, 1962; Livingstone and Hubel, 1984; Albrecht and Hamilton, 1982; Gallant et al., 1993; Hubel and Wiesel, 1968; Desimone et al., 1984; Tanaka et al., 1991; Pasupathy and Connor, 2002; Yue et al., 2020; Yang et al., 2023; Pasupathy and Connor, 1999; Tsunoda et al., 2001; Rust and DiCarlo, 2010; Brincat and Connor, 2004; Zeki, 1973; Pasupathy and Connor, 2001; Moran and Desimone, 1985; Kobatake and Tanaka, 1994; Kriegeskorte et al., 2008; Kobatake et al., 1998; Miyashita, 1988).

The advent of deep convolutional neural networks (DCNNs) marked a significant improvement in predictive accuracy across the visual system (Yamins et al., 2014; Abdelhack and Kamitani, 2018; Wen et al., 2018; Horikawa and Kamitani, 2017; Eickenberg et al., 2017; Güçlü and Van Gerven,

2015; Cichy et al., 2016; Khaligh-Razavi and Kriegeskorte, 2014; Schrimpf et al., 2020; Storrs et al., 2021; Safarani et al., 2021; Schwartz et al., 2019; Seeliger et al., 2021). DCNNs trained on image categorization tasks emerged as the first class of models capable of capturing neural activity in the primate visual cortex with a reasonable degree of fidelity. This success spurred a wave of model-brain comparisons, wherein variations in input data, architecture, and learning objectives were explored to identify the most predictive models of brain responses in both non-human primates and humans.

More recently, models trained using multimodal contrastive learning approaches, such as CLIP, or image-caption embeddings from large language models (LLMs), have shown promise in predicting neural responses in the visual cortex (Tang et al., 2024; Wang et al., 2022; Doerig et al., 2024). These findings suggest that visual brain responses may encode some linguistically learned structure or semantics. In parallel, another class of models, optimized specifically for neural response prediction (Khosla and Wehbe, 2022; Khosla et al., 2022; Federer et al., 2020; Dapello et al., 2022; St-Yves et al., 2023) — either trained from scratch or fine-tuned to better align with primate visual representations—has achieved impressive predictive accuracy, particularly with the availability of large-scale neural datasets.

Given the broad range of modeling approaches applied to different regions of the visual cortex, a critical question remains: which approach offers the most quantitatively precise predictions of neural responses across the various areas of the human visual system? This challenge underscores the need for systematic comparisons to determine the optimal models for different visual processing stages. While some recent studies have made strides in conducting large-scale comparative analyses, they tend to focus primarily on specific pre-selected visual regions and largely compare different task-optimized vision networks (Conwell et al., 2022b). A more comprehensive comparison is needed to evaluate a broader set of approaches, including models based on response optimization and embeddings from language models trained on vision-aligned tasks or pure language data.

A related but less explored area in human visual research involves investigating different readout mechanisms for predicting neural responses from neural representations. The predominant readout in primate studies is the fully-connected affine readout, often used in regularized linear regression models. However, these linear ridge regression readouts require numerous parameters, especially in high-dimensional spaces, leading to significant computational and memory demands. To mitigate this, more efficient methods have been developed, such as factorized linear readouts by (Klindt et al., 2017), that decouple spatial from feature selectivity, reducing overhead and improving prediction accuracy. The Gaussian2D readout (Lurz et al., 2020) further enhances parameter efficiency by learning spatial readout locations using a bivariate Gaussian distribution informed by anatomical retinotopy. Given this diversity, a key question is which readout method best predicts neural responses across different regions of the human visual cortex, and how can these methods be refined for better predictions? Addressing these questions is essential for improving neural response models and computational modeling frameworks.

To address the above challenges, we make the following key contributions in this paper:

1. **Introduction of a novel readout mechanism:** We introduce a novel readout method utilizing Spatial Transformers, which delivers significant improvements in accuracy (3-23%) compared to previously employed SOTA readouts.
2. **Identification of brain regions responsive to visual and semantic input:** Through comparative analysis of models across various visual regions, we identify three distinct regions in the human visual cortex that respond primarily to (a) perceptual characteristics of the input, (b) localized visual semantics aligned with linguistic descriptions, and (c) global semantic interpretations of the input, also aligned with language.
3. **Comprehensive analysis of neural network models:** We conduct an in-depth analysis of various artificial neural network models, incorporating both vision and language inputs. Additionally, we explore different readout mechanisms and examine which models perform better in specific brain regions, while highlighting the unique advantages each provides.

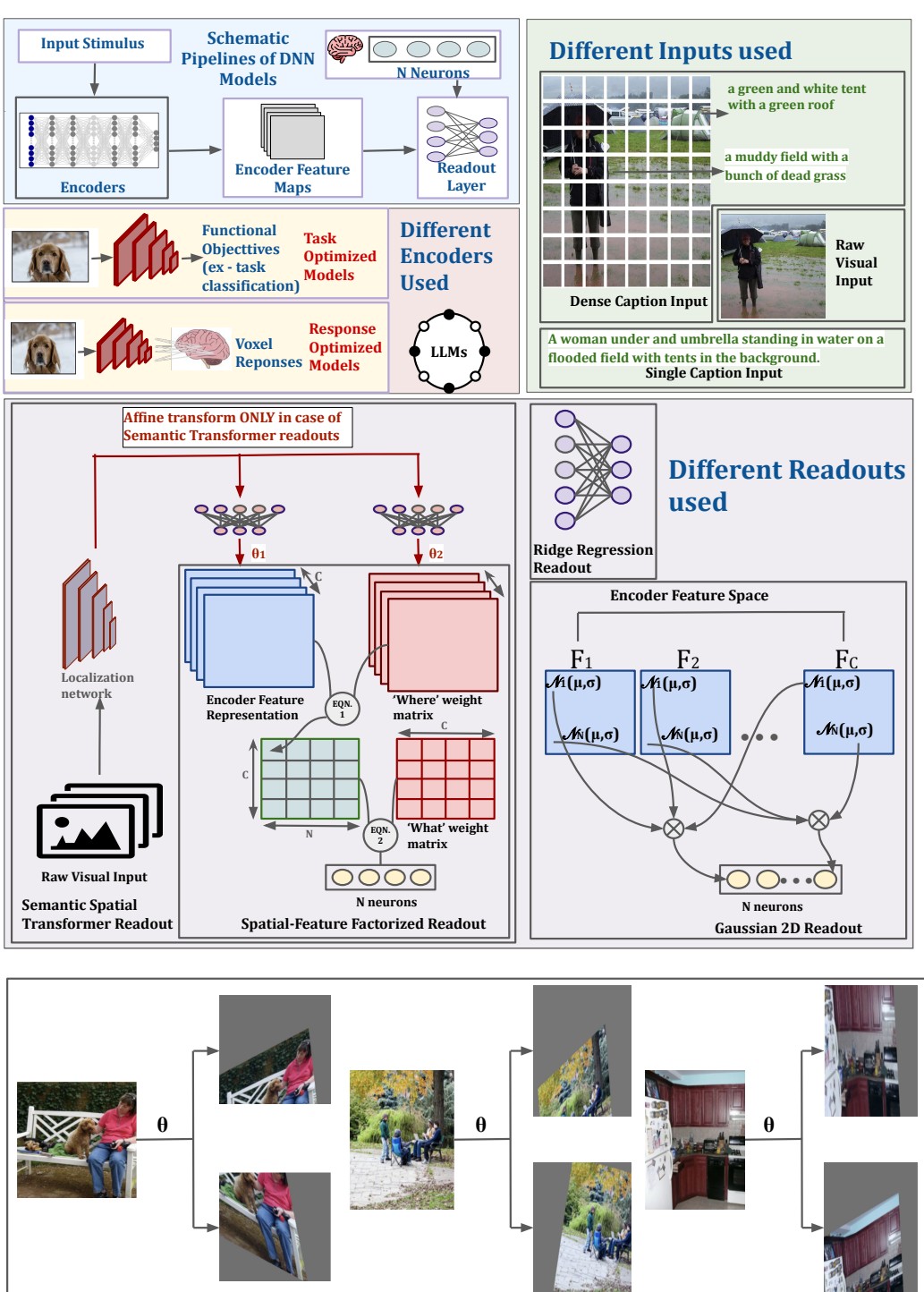

Figure 1: Above - Overall Schematic of the models - the various inputs to the models (Visual and Language) and the four kinds of readout mechanisms (Linear, Gaussian, Factorized and Semantic Spatial Transformer), Below - The learned transformations $\theta_1$ for specific channels of the encoder's feature representation are applied to the input image to demonstrate how each respective feature map is modulated as an example.

## 2 METHODS

### 2.1 ENCODERS

**Task Optimised Models** - We use encoders from pre-trained models like AlexNet and ResNet, originally trained for object classification on the large-scale ImageNet dataset. The weights of their intermediate layers are frozen, and only the readout layers (described later) are trained. Prior research shows that early layers of neural networks align with lower visual cortex regions, while later layers correspond to higher regions (Khaligh-Razavi and Kriegeskorte, 2014; Güçlü and Van Gerven, 2015; Cichy et al., 2016; Eickenberg et al., 2017; Horikawa and Kamitani, 2017; Wen et al., 2018; Abdelhack and Kamitani, 2018; Yamins et al., 2014). Thus, we experimented with all layers of task-optimized networks. For fair comparison, we selected the best-performing layers for each cortical region (see Appendix Table 2 and summary in Table 1).

**Response Optimised Models** - Task-optimized models often rely heavily on established a priori hypotheses, which may be biased towards pre-existing conclusions, limiting novel discoveries. Further, these networks are typically optimized for specific tasks, such as object classification, which may not capture the full range of visual processing in the cortex. Recently, (Khosla and Wehbe, 2022) showed that training neural networks from scratch with stimulus images and fMRI data from the NSD dataset (Allen et al., 2022) can achieve accuracy comparable to SOTA task-optimized models. By training directly on neural data, the network is not constrained by biases introduced from tasks like object recognition (as in predominant task-optimized networks), allowing it to learn more flexible or diverse representations that are more specific to the neural responses across different brain regions or stimuli.

We leverage the same architecture for response-optimized models as prior work (Khosla and Wehbe, 2022), which consists of a convolutional neural network (CNN) core that transforms raw input data into feature spaces characteristic of different brain regions, followed by a readout layer that maps these features to fMRI voxel responses (Figure 1-A). The core contains four convolutional blocks, where each convolutional block includes two convolutional layers, followed by internal batch normalization, nonlinear ReLU activations, and an anti-aliased average pooling operation. To ensure equivariance under all isometries, we use E(2)-Equivariant Steerable Convolution layers (Weiler and Cesa, 2019). Further analysis on the importance of network architecture for Response Optimized models can be found in Supplementary Section A.6 and Table 7.

**Language Models** - Recent studies show that higher visual regions converge toward representational formats similar to large language model (LLM) embeddings of scene descriptions. (Doerig et al., 2024) used MPNET (Song et al., 2020) to encode image captions and map them to fMRI responses via ridge regression, finding it effectively modeled higher visual areas despite being trained on language inputs alone. In contrast, (Tang et al., 2024) and (Wang et al., 2022) used multimodal models like CLIP (Radford et al., 2021) and BridgeTower (Yang et al., 2023), and showed that CLIP outperforms vision-only models in capturing higher regions due to language feedback. The language models preserve ROIs' category selectivities, and were more effective at predicting responses to visual data compared to the reverse. These insights motivated us to evaluate language models relative to vision-only response-optimized and task-optimized models as detailed below (A more detailed comparison on CLIP and MPNET embeddings can be found in Appendix A.2) -

1. **Single Caption** - Images in the NSD dataset are sourced from MS COCO (Lin et al., 2014) and annotated by 4-5 human annotators. We encode these captions using CLIP or MPNET, average the encodings, and input them into a one-layer linear regressor to map them to fMRI voxel responses. Since the captions describe the image as a whole without offering spatial details (i.e., fine-grained delineations of features at different locations), we only use the ridge linear readout for single caption inputs as shown in Figure 1-A.

2. **Dense Caption** - An image of size $424 * 424$ is divided into grids of size $53 * 53$. For each grid, a caption is generated using GPT-2, which is then encoded by either CLIP or MPNET. Thus an image of shape $3 * 424 * 424$ is transformed into a feature representation $N * 8 * 8$, where N is the size of the embedding produced by CLIP or MPNET. The dense-caption language encoders further process these feature maps through a single convolutional block (as described earlier for the response-optimized vision encoders) before passing them to the readout model. More details can be found at Supplementary Section A.3 and Figure 9.

## 2.2 READOUTS

The encoders discussed above are paired with a readout model (Figure 1-A) that maps the encoder feature representations to voxel fMRI responses from various regions of the visual cortex.

**Linear Readout** consists of a ridge regression model that takes in the flattened feature representations from the encoders as inputs and maps them to voxel fMRI representations. Specifically, for a given stimuli $i$, the response vector $\boldsymbol{Y}_i \in \mathbb{R}^n$ (where $n$ represents the number of voxels) is predicted as $\boldsymbol{Y}_i = \hat{\boldsymbol{W}} \boldsymbol{E}_i$, where $\boldsymbol{E}_i \in \mathbb{R}^e$ refers to the flattened feature representation from the encoder and $\hat{\boldsymbol{W}} \in \mathbb{R}^{n*e}$ refers to the linear readout weights. The weights $\hat{\boldsymbol{W}}$ are obtained by minimizing the ridge regression objective function: $f(\boldsymbol{W}) = \min_{\boldsymbol{W}} ||\boldsymbol{Y} - \boldsymbol{E}\boldsymbol{W}||_F^2 + \lambda||\boldsymbol{W}||_F^2$, where $\lambda$ is the regularization parameter. We find the optimal $\lambda$ through cross-validation.

**Gaussian 2D Readout** (Lurz et al., 2020) models each neuron's spatial sensitivity as a 2D Gaussian in the input feature space. Each voxel is assumed to respond to a specific region of the input, with sensitivity characterized by a bivariate normal distribution. The mean represents the neuron's most sensitive location (receptive field center), while the covariance matrix determines the size, shape, and orientation of the receptive field along the $x$ and $y$ axes. This Gaussian is applied uniformly across all input channels, indicating shared spatial sensitivity. Neuron responses are computed by bilinearly interpolating values from each channel based on the Gaussian distribution, weighting them by learned channel-specific weights, and summing the contributions to get the final voxel response.

For a voxel $n$ with receptive field defined by $G_n(x, y) \sim \mathcal{N}(\mu_n, \Sigma_n)$, the voxel response $\boldsymbol{Y}_n$ is calculated as: $\boldsymbol{Y}_n = \Sigma_c \boldsymbol{W}_{nc} \boldsymbol{V}_c(x, y)$, where $\boldsymbol{W}_{nc}$ is the learned weight that modulates the contribution of channel $c$ to voxel $n$, $\boldsymbol{V}_c(x, y)$ is the feature value sampled from channel $c$ of the encoder feature map $\boldsymbol{E} \in \mathbb{R}^{C \times W \times H}$ at location $(x, y)$, using bilinear interpolation based on the receptive field defined by $G_n(x, y)$, and $C$ is the number of channels in the encoder output.

**Spatial-Feature Factorized Linear Readout** factorizes the linear readout model into spatial (the portion of the input space a voxel is sensitive to) and feature (the specific features of the input space a voxel responds to) dimensions, as described in (Klindt et al., 2017). By separating spatial (where) and feature (what) dimensions, the model mirrors the known structure of neural receptive fields in the brain, where neurons exhibit sensitivity to specific spatial locations and particular feature types. This approach not only significantly reduces the number of parameters but also aligns more closely with the known characteristics of neural responses.

$$\boldsymbol{Y}_{c,n} = \Sigma_w \Sigma_h \boldsymbol{E}_{c,w,h} \boldsymbol{S}_{n,w,h} \tag{1}$$
$$\boldsymbol{Y}_n = \Sigma_c \boldsymbol{Y}_{c,n} \boldsymbol{F}_{n,c} \tag{2}$$

where $\boldsymbol{Y} \in \mathbb{R}^n$ comprises responses of $n$ voxels, $\boldsymbol{E} \in \mathbb{R}^{C*W*H}$ is the feature representation (what) from the encoder, $\boldsymbol{S} \in \mathbb{R}^{N*W*H}$ refers to the spatial (where) weights and $\boldsymbol{F} \in \mathbb{R}^{N*C}$ refer to the feature weights with $N$ = number of voxels in the brain region, $C$ = number of channels in the encoder feature representation and $W, H$ being the width and height of the encoder feature representation respectively.

**Semantic Spatial Transformer Readout** - We introduce a novel readout that modulates the encoder feature representation and spatial weights for every voxel depending on the stimulus using a learnable module based on the Spatial Transformer Network (STN) (Jaderberg et al., 2015). The STN in essence allows us to 'zoom in' on relevant portions of the image (e.g., via rotation, scaling, translation, or affine transformations) (Figure 1-B) and these relevant regions are determined independently for every voxel and for every channel. By learning to apply the right transformations, STNs adjust inputs to a canonical form, effectively making the model invariant to geometric transformations.

This module comprises four elements - (1) A localization network (a pretrained resnet50 model) which takes in the image as input and returns a feature representation of the image (before the adaptive average pooling operation), (2) Two linear deformation networks that take in the feature representation from the localization network and return two sets of affine transformations $\theta_1 \in \mathbb{R}^{C*6}$ and $\theta_2 \in \mathbb{R}^{N*6}$ as shown in Figure 1-A, where $C$ is the number of channels of the encoder feature representation and $N$ is the number of voxels, (3) A parameterised sampling grid that generates a transformed grid of the same shape as the encoder feature representation $\boldsymbol{E}$ using $\theta_1$ and of the same shape as the spatial weight matrix $\boldsymbol{S}$ using $\theta_2$ and (4) A sampler that transforms $\boldsymbol{E}$ and $\boldsymbol{S}$

with the above grid via bilinear interpolation. Here, $Y_n$ is calculated in the same way as in the Spatial-Feature Factorized Linear Readout (Equations 1, 2), with $E$ replaced by $E' = AT(E, \theta_1)$ and $S$ replaced by $S' = AT(S, \theta_2)$. $AT(X, \theta)$ with $X \in \mathbb{R}^{M*W*H}$ and $\theta \in \mathbb{R}^{M*2*3}$ performs an affine transformation $\theta_m$ to each channel $m$ in $X$.

Different channels in a feature map often encode distinct attributes, such as edges, textures, or shapes. By allowing channel-specific transformations, the STN can adapt to the unique geometric properties of the features represented in each channel. Unlike object classification tasks, where we can employ augmentations tailored to known invariances (e.g. rotating an image won't change the category label) to boost predictive accuracy, the geometric invariances of voxel responses are unknown. STN enables the network to learn these invariances directly from the data, providing a crucial advantage in predicting voxel responses across diverse visual regions. Moreover, STN also allows voxel-specific spatial modulation of receptive fields (RFs), inspired by studies demonstrating the dynamic adaptability of RFs to stimulus properties - RF sizes can expand or contract based on contrast Sceniak et al. (1999) and can also shift or reshape in response to contextual or attentional influences Womelsdorf et al. (2006). Unlike fixed spatial masks used in the previous readouts, the STN employs affine transformations to capture stimulus-dependent spatial changes such as translation, scaling, rotation, and shearing. This flexibility enables more accurate modeling of neural responses that exhibit dynamic RF properties. Further analysis on this readout is expanded in Supplementary Table 5, Figure 10 and Section A.4.

## 2.3 TRAINING AND DATASET

In this study, we utilized stimuli-response pairs from four subjects (Subjects 1, 2, 5, and 7) from the Natural Scenes Dataset (More details in Appendix A.1). The experimental setup involved presenting a total of 37,000 image stimuli from the MS COCO dataset (Lin et al., 2014) to these subjects. Out of these, 1,000 images were shown to all four subjects, and these shared images were designated as the test set for our analyses. The remaining 36,000 images were split into 35,000 for training and 1,000 for validation purposes. We trained separate models for each of the following brain regions: the high-level ventral, dorsal and lateral streams, V4, V3v, V3d, V2v, V2d, V1v, and V1d. This approach allowed us to tailor the models to the unique neural response patterns of each region, thereby providing a more precise understanding of how different parts of the visual cortex process information. Throughout the paper, the reported accuracy refers to the test-time performance, measured as the noise-normalized Pearson correlation between predicted and actual voxel responses.

All models were trained using an NVIDIA GeForce RTX 4090 and NVIDIA A40 GPU. We employed a batch size of 4 with gradient accumulation to achieve an effective batch size of 16, using a learning rate of 0.0001. Training was performed using an equal-weighted combination of Mean Squared Error (MSE) and correlation loss between predicted and target voxel responses, with early stopping applied after 20 epochs without improvement in validation accuracy, measured by Pearson correlation.

## 3 RESULTS

### 3.1 PERFORMANCE COMPARISON OF READOUTS ACROSS VISION AND LANGUAGE MODELS IN THE VISUAL CORTEX

We first evaluated the performance of various readout mechanisms in predicting neural responses across different brain regions. Our results showed that the Semantic Spatial Transformer readouts consistently outperform linear, 2D Gaussian, and Spatial-Feature Factorized Linear readouts across all regions of the visual cortex and for almost all encoder models (see Figure 2). This is due to the Spatial Transformer Network (STN) in the readout, which applies learnable transformations to encoder feature maps, converting them into canonical forms. This process removes irrelevant geometric variations while emphasizing key feature characteristics. Additionally, the STN learns transformations for spatial weights, enabling stimulus-dependent modulation of receptive field maps for each voxel, providing greater flexibility in modeling complex inputs. This trend of superior performance is especially evident in vision models (see Figure 2-A) and holds for other task-optimized encoders processing visual input (details in Appendix Tables 2 and 3). Figure 2-B further illustrates the brain voxels where each readout performs best, underscoring the dominant performance of the Semantic Spatial Transformer readout for vision models.

While the Semantic Spatial Transformer achieves the overall highest accuracy across all regions for all models (Appendix Tables 2, 3, 4), its improvement is less pronounced with language embedding inputs (Figure 2-B). This disparity arises because the Semantic Spatial Transformer readout uses a pretrained ResNet50 encoder as the localization network to learn affine transformations that adjust both vision and language encoder feature spaces. Vision encoder features are generally larger per channel (e.g., 28×28) than language encoder features (e.g., 4×4). Consequently, the Semantic Spatial Transformer readout has a greater capacity to leverage the rich spatial information available in vision models. Larger spatial dimensions provide more granular information, allowing STNs to learn transformations that account for variations in position, scale, and orientation of features more accurately. Further analysis on bias introduced by readouts can be found in Supplementary Section A.5 and Table 6.

Further, Spatial-Feature Factorized Linear Readouts outperform linear ridge regression readouts both in terms of memory efficiency and prediction performance, as shown in Figure 2-A and Appendix Tables 4, 3 and 2. This improvement is attributed to the readout's capability to effectively disentangle voxel response selectivity into spatial and feature dimensions. This approach aligns with established phenomena in neuroscience, where neurons exhibit selectivity not only for specific features but also for stimuli presented within their receptive field locations.

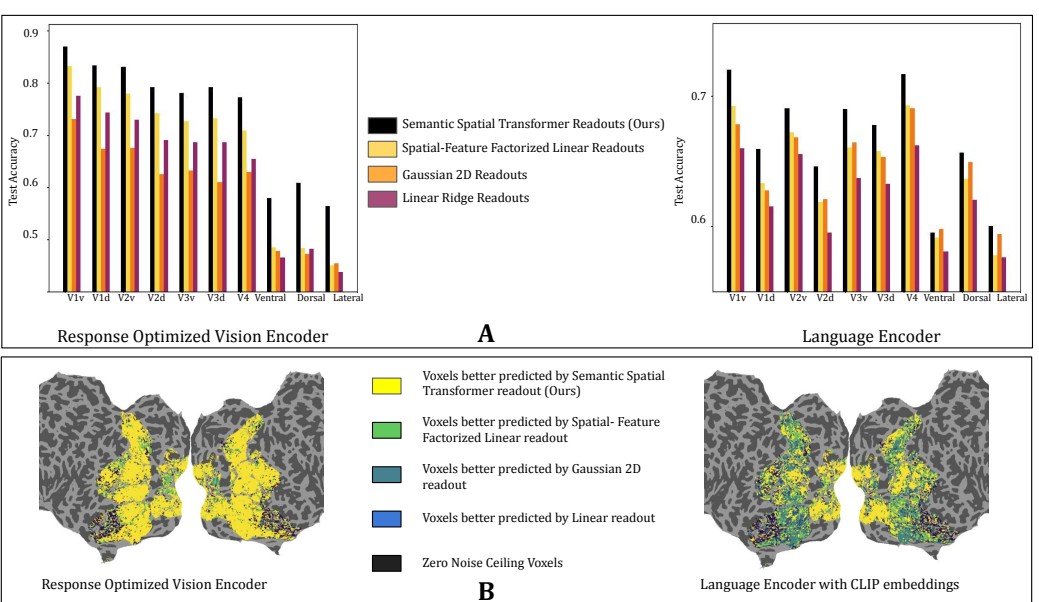

Figure 2: Comparison of various readout mechanisms - (A) Noise Normalized Test Accuracy (Pearson Correlation) on held out dataset for different brain regions calculated using Response optimized Vision and Dense Language (CLIP embedding) models using four different readouts , (B) Brain Visualizations showing regions where each readout performs the best

Bivariate Gaussian readouts are mostly outperformed by both spatial-feature factorized linear readouts and linear ridge regression readouts in vision models, despite needing significantly fewer parameters. This performance gap can be attributed to the fact that Gaussian readouts were initially developed for grayscale stimuli in the mouse primary visual cortex (Lurz et al., 2020), where they utilized the brain's retinotopic mapping and anatomical organization to accurately define receptive fields. In our study, however, we learn the parameters of the Gaussian readout solely from the responses to complex image inputs, deliberately excluding anatomical information to maintain a fair comparison with other methods. Furthermore, this modeling approach may be less effective for the human visual system, where the assumption of a Gaussian-like structure may not hold true for the spatial receptive fields of all voxels, which may exhibit greater complexity.

Interestingly, the performance gap between Gaussian readouts and other readouts narrows in language models, where Gaussian readouts slightly outperform linear readouts across all regions and exceed Spatial-Feature Factorized Linear Readouts in higher regions. This may be due to the smaller feature

space in language models compared to vision models (e.g., 4×4 vs. 28×28), which simplifies receptive field localization.

As the Semantic Spatial Transformer readouts outperformed others across all regions and models, we will focus on this readout when analyzing the encoders in detail in the following sections.

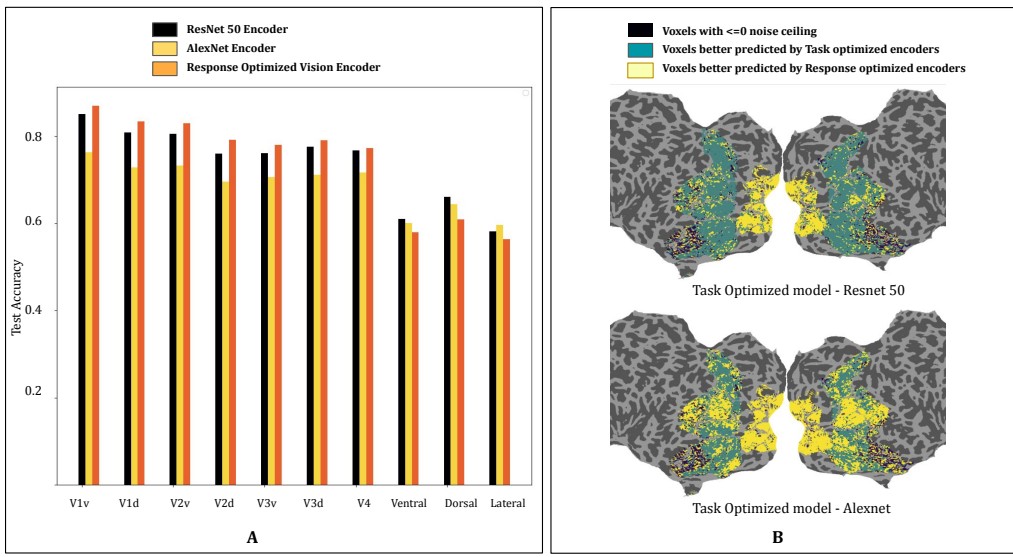

Figure 3: Comparison of Task optimized Vision models versus Response Optimized Vision models - (A) Test Accuracy (Normalized Pearson Correlation) on held out dataset using Task optimized model encoders and Response Optimized model encoders with Semantic Spatial Transformer readout, (B) Brain Visualization showing regions better predicted by each model

| Model Details | V1v | V1d | V2v | V2d | V3v | V3d | V4 | Ventral | Dorsal | Lateral |
|---|---|---|---|---|---|---|---|---|---|---|
| TV | 0.8507 | 0.8083 | 0.8057 | 0.7603 | 0.7612 | 0.7763 | 0.7674 | **0.6105** | **0.6606** | 0.5823 |
| RV | **0.8698** | **0.8340** | **0.8302** | **0.7919** | **0.7808** | **0.7913** | **0.7729** | 0.5796 | 0.6089 | 0.5638 |
| SL | 0.3974 | 0.3779 | 0.3809 | 0.3702 | 0.4093 | 0.4119 | 0.4882 | 0.5661 | 0.6243 | 0.5920 |
| DL | 0.7196 | 0.6590 | 0.6903 | 0.6457 | 0.6897 | 0.6774 | 0.7167 | 0.5953 | 0.6562 | **0.6001** |

Table 1: Performance (Test Accuracies as Noise Normalized Pearson Correlation) of Task Optimized Vision models (ResNet 50) - TV (best results from 2), Response Optimized Vision models - RV and Language Models (with CLIP embeddings), both Single Captioned (SL) and Dense Captioned (DL) all with Semantic Spatial Transformer readout (except Single Caption models with Ridge Linear reasout)

## 3.2 Vision: Task Optimised Models Vs Response Optimised Models

To ensure a fair comparison, we trained models using different sets of layers for each task-optimized model (Appendix Table 2), and used only the best-performing ResNet50 layers for comparison, as presented in Table 1. In the early regions of the visual cortex (V1, V2, V3, and V4), response-optimized vision models consistently outperform task-optimized models by 2-12% (Figure 3 and Table 1), with a particularly notable margin over simpler architectures like AlexNet (Appendix Table 2). This suggests that features necessary for modeling early and mid-level visual areas are not fully captured by current task-optimized models, and explicit alignment with neural responses is crucial for higher prediction accuracy. This may be because task-optimized models, primarily trained on object-centric tasks, don't account for the broader range of visual functions performed by the brain. Incorporating more ethologically relevant tasks into the optimization framework might be necessary for better modeling of early to mid-level visual processing. In the higher regions of the visual cortex (high-level ventral, dorsal, and lateral streams), task-optimized models show a slight performance advantage of around 5% over response-optimized models. This could be because these regions

process more complex visual information, and task-optimized models, trained on larger object-centric datasets like ImageNet ($\geq$1.2 million images), better capture these functions. However, the small difference indicates that response-optimized models, despite being trained on only a fraction ( 3%) of the data, still capture significant aspects of high-level visual processing.

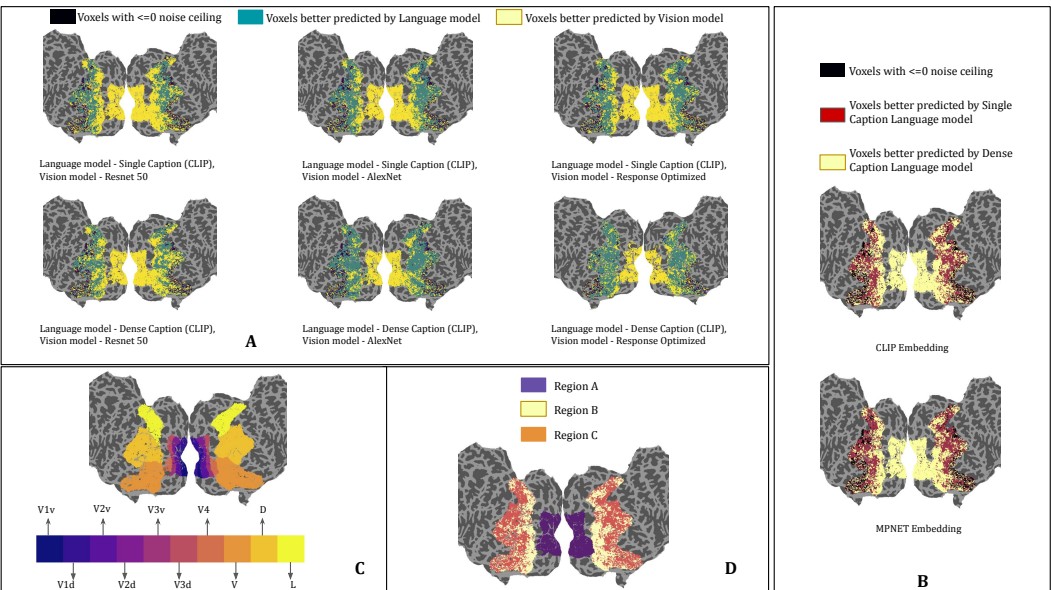

Figure 4: Comparison of Vision and Language models using Semantic Spatial Transformer readouts - (A) Brain Visualizations showing voxels that are better predicted by vision models and language models, (B) Brain Visualizations showing voxels that are better predicted by single caption and dense caption language models, (C) Brain Visualizations showing the ten regions of the human visual cortex analysed in this study (V, D and L refer to Ventral, Dorsal and Lateral streams respectively), (D) Brain Visualizations highlighting three distinct regions, each demonstrating varying sensitivities to largely perceptual characteristics of the input, localized visual semantics aligned with linguistic descriptions, and global semantic interpretations of the input, also aligned with language

### 3.3 BRAIN REGIONS SENSITIVE TO VISION VS LANGUAGE MODELS

Recent research shows that pure language models, like MPNET, can predict image-evoked brain activity in the high-level visual cortex using only image captions (Doerig et al., 2024). This raises intriguing questions about the alignment between the human visual cortex and language. To explore this relationship further, we compare these language models with vision-only models.

When we assess language models that receive only image captions—without the images themselves—against response-optimized vision models, we find that the lower regions of the visual cortex are better modeled by vision-based approaches. In contrast, higher regions are more effectively captured by language models (see Figure 4-A, column 3 and Table 1). This pattern also holds when comparing language models to task-optimized vision models, although the distinction is less pronounced (first two columns of Figure 4-A).

Next, we differentiate between single-caption and dense-caption models. Single-caption models convey only the overall semantic content of an image, whereas dense-caption models capture both spatial and semantic details. Consequently, the lower regions of the visual cortex, which are sensitive to fine-grained visual information, are better modeled by dense-caption models, as illustrated in Figure 4-B.

As we move from the lower to the higher regions of the visual cortex, there is a notable shift in sensitivity from localized semantics to global semantics across all ventral, dorsal, and lateral streams. Figure 4-B demonstrates that single-caption models dominate in the mid-to-higher regions of these streams, emphasizing the sensitivity of these areas to the overall meaning or interpretation of an entire

image or scene. This trend is further corroborated in Figure 4-A, which compares vision models with both single-caption and dense-caption language models. Here, response-optimized vision models outperform single-caption models in the lower regions of the ventral, dorsal, and lateral streams, but do not maintain this advantage in the mid-to-higher regions.

Thus, we can identify three distinct regions in the visual cortex that are sensitive to different types of stimulus (Figure 4-D): (1) the lower visual regions (V1, V2, V3, and V4) are most sensitive to perceptual features that are not fully captured by linguistic descriptions - region A; (2) the mid-level regions of the dorsal, ventral, and lateral streams are most sensitive to localized semantics (i.e. detailed, specific information about particular parts or regions of an image) - region B; and (3) the higher regions of the dorsal, ventral, and lateral streams are sensitive exclusively to global semantic information - region C. Vision models outperform both single and dense caption language models in region A (Figure 4-A and Table 1), thus proving its sensitivity to largely perceptual features. Dense Caption language models outperform single caption language models (Figure 4-B) and response-optimized vision models (Figure 4-A) in region B, thus proving it is most sensitive to nuanced, localized semantic details. Vision models also outperform single caption models in region B (Figure 4-A), thus proving it is more sensitive to detailed visual information. Lastly, single caption language models outperform both dense caption models (Figure 4-B) and vision models (Figure 4-A) in region C, thus confirming its sensitivity to global semantics. Although this comparison was done extensively using Semantic Spatial Transformer readout, the trends hold true for other readouts as well, although to a much lesser extent (Appendix Figures 7, 6, 5).

## 4 DISCUSSION

In this study, we leveraged the Natural Scenes Dataset to evaluate various neural network models in predicting neural responses across different brain regions. Our analysis focused on three key comparisons: task-optimized vs. response-optimized models, vision models vs. language models, and different readout methods for mapping model activations to brain signals.

First, we compared task-optimized models, pre-trained on specific visual tasks and thus biased toward those tasks, with response-optimized models trained directly from brain response data. Our results show that response-optimized models, which learn from raw visual inputs, significantly outperform task-optimized models in early visual regions. This suggests that brain-like processing in early-to-mid visual areas does not fully emerge in task-optimized models, and explicit alignment with neural data enhances prediction accuracy. However, in higher visual regions, both model types perform comparably, with task-optimized models showing a slight edge.

Next, we compared vision models with language models, including single-caption and dense-caption models. Vision models outperformed language models in early visual regions, which are more attuned to perceptual features not captured by linguistic descriptions. In mid-level visual regions, sensitivity shifts toward semantic information, with dense-caption models excelling due to their ability to represent localized semantics. In higher visual regions, single-caption models perform better, indicating the importance of global scene understanding.

Finally, we evaluated different readout mechanisms for mapping activations to brain responses. Factorized readouts significantly outperformed linear models, and incorporating a Semantic Spatial Transformer further improved performance, particularly in vision models.

Our work has several limitations. First, we focused on task-optimized models trained for object categorization. A comprehensive comparison of models trained on other visual objectives and data sets is outside the scope of this study. However, prior research suggests that variations in architecture, objective, and data diet do not drastically impact response prediction accuracy, so we do not expect our conclusions to change significantly with additional models. While we found that language models become more accurate in predicting responses in high-level visual regions, we did not explore what specifically drives this performance. It is still uncertain whether object category information (e.g., nouns) or other elements such as actions, spatial relationships, or contextual details play a more significant role. Finally, while the Semantic Spatial Transformer led to better predictions, future work should investigate how spatial and feature weights are modulated by different inputs. We also only tested affine transformations; more constrained or nonlinear deformations may offer further improvements.

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

## A APPENDIX

| Model Details | | Visual Cortex Region | | | | | | | | | |
|---|---|---|---|---|---|---|---|---|---|---|---|
| Layers | Readout | V1v | V1d | V2v | V2d | V3v | V3d | V4 | Ventral | Dorsal | Lateral |
| | | ResNet 50 | | | | | | | | | |
| 1 | R | 0.6009 | 0.5695 | 0.5168 | 0.4783 | 0.4612 | 0.4543 | 0.4085 | 0.2958 | 0.3101 | 0.2508 |
| | G | 0.5935 | 0.5634 | 0.5110 | 0.4238 | 0.4135 | 0.4148 | 0.3758 | 0.2236 | 0.1928 | 0.1940 |
| | F | 0.8041 | 0.7627 | 0.7321 | 0.6950 | 0.6517 | 0.6540 | 0.5771 | 0.3252 | 0.3318 | 0.2709 |
| | S (Ours) | 0.8498 | 0.8022 | 0.7860 | 0.7501 | 0.7559 | 0.7461 | 0.7410 | 0.5763 | 0.6208 | 0.5652 |
| 2 | R | 0.5618 | 0.6535 | 0.6276 | 0.4677 | 0.4564 | 0.4485 | 0.4157 | 0.3628 | 0.3796 | 0.3131 |
| | G | 0.6478 | 0.5827 | 0.5694 | 0.5086 | 0.4975 | 0.4861 | 0.4898 | 0.2958 | 0.2797 | 0.2593 |
| | F | 0.8142 | 0.7728 | 0.7601 | 0.7302 | 0.6956 | 0.7110 | 0.6403 | 0.4034 | 0.4116 | 0.3515 |
| | S (Ours) | **0.8507** | **0.8083** | **0.8057** | **0.7603** | **0.7612** | **0.7763** | 0.7601 | 0.5813 | 0.6241 | 0.5667 |
| 3 | R | 0.6599 | 0.6413 | 0.6426 | 0.6014 | 0.6051 | 0.6237 | 0.6138 | 0.5022 | 0.5657 | 0.4689 |
| | G | 0.6607 | 0.6110 | 0.6359 | 0.5920 | 0.6270 | 0.6205 | 0.6526 | 0.4991 | 0.5296 | 0.4671 |
| | F | 0.8046 | 0.7666 | 0.7705 | 0.7482 | 0.7465 | 0.7675 | 0.7540 | 0.5751 | 0.6277 | 0.5384 |
| | S (Ours) | 0.7898 | 0.7393 | 0.7643 | 0.7193 | 0.7496 | 0.7495 | **0.7674** | **0.6105** | **0.6606** | **0.5823** |
| 4 (all) | R | 0.2812 | 0.2577 | 0.2583 | 0.2556 | 0.2880 | 0.2433 | 0.3132 | 0.3006 | 0.2922 | 0.2820 |
| | G | 0.5170 | 0.4671 | 0.4810 | 0.4318 | 0.4821 | 0.4787 | 0.5442 | 0.4764 | 0.4702 | 0.4704 |
| | F | 0.5922 | 0.5488 | 0.5606 | 0.5297 | 0.5542 | 0.5659 | 0.5612 | 0.4525 | 0.4741 | 0.4269 |
| | S (Ours) | 0.6989 | 0.6504 | 0.6746 | 0.6487 | 0.6743 | 0.6791 | 0.6814 | 0.5857 | 0.6337 | 0.5809 |
| | | AlexNet | | | | | | | | | |
| 1 | R | 0.6359 | 0.6320 | 0.5844 | 0.5403 | 0.5268 | 0.5178 | 0.4795 | 0.3134 | 0.3275 | 0.2727 |
| | G | 0.6520 | 0.6009 | 0.5539 | 0.5197 | 0.4649 | 0.4489 | 0.4550 | 0.3156 | 0.3054 | 0.2662 |
| | F | 0.7253 | 0.6897 | 0.6479 | 0.6136 | 0.5678 | 0.5841 | 0.5300 | 0.3170 | 0.3178 | 0.2763 |
| | S (Ours) | 0.7590 | 0.7159 | 0.7229 | 0.6662 | 0.6934 | 0.6764 | 0.7004 | 0.5594 | 0.6072 | 0.5556 |
| 2 | R | 0.5822 | 0.5550 | 0.5268 | 0.4951 | 0.4919 | 0.4855 | 0.4715 | 0.2924 | 0.2949 | 0.2485 |
| | G | 0.6459 | 0.6221 | 0.5883 | 0.5489 | 0.5439 | 0.5357 | 0.5278 | 0.3688 | 0.3399 | 0.3271 |
| | F | 0.7325 | 0.6923 | 0.6704 | 0.6396 | 0.6168 | 0.6287 | 0.5876 | 0.3864 | 0.3822 | 0.3322 |
| | S (Ours) | 0.7710 | **0.7288** | 0.7273 | 0.6950 | 0.7043 | **0.7117** | **0.7169** | 0.5705 | 0.6002 | 0.5408 |
| 3 | R | 0.5951 | 0.5722 | 0.5554 | 0.5260 | 0.5234 | 0.5313 | 0.5197 | 0.3389 | 0.3392 | 0.2879 |
| | G | 0.6311 | 0.6150 | 0.5997 | 0.5705 | 0.5597 | 0.5629 | 0.5719 | 0.4289 | 0.4072 | 0.3888 |
| | F | 0.7419 | 0.7055 | 0.7033 | 0.6713 | 0.6655 | 0.6864 | 0.6594 | 0.4618 | 0.4711 | 0.4098 |
| | S (Ours) | **0.7634** | 0.7236 | **0.7327** | **0.6961** | **0.7065** | 0.7092 | 0.7148 | 0.5694 | 0.6071 | 0.5326 |
| 4 | R | 0.6145 | 0.5830 | 0.5834 | 0.5527 | 0.5733 | 0.5677 | 0.5632 | 0.4123 | 0.4181 | 0.3486 |
| | G | 0.6357 | 0.6038 | 0.5994 | 0.5677 | 0.5684 | 0.5795 | 0.5908 | 0.4601 | 0.4827 | 0.4220 |
| | F | 0.7325 | 0.6933 | 0.6989 | 0.6724 | 0.6735 | 0.6895 | 0.6758 | 0.5066 | 0.5323 | 0.4559 |
| | S (Ours) | 0.7444 | 0.7070 | 0.7173 | 0.6816 | 0.7037 | 0.7108 | 0.7129 | 0.5688 | 0.6214 | 0.5458 |
| 5 (all) | R | 0.4931 | 0.4798 | 0.4703 | 0.4474 | 0.4560 | 0.4609 | 0.4662 | 0.3717 | 0.3806 | 0.3394 |
| | G | 0.5605 | 0.5652 | 0.5136 | 0.5193 | 0.4946 | 0.5231 | 0.5260 | 0.4523 | 0.4334 | 0.4201 |
| | F | 0.6889 | 0.6339 | 0.6679 | 0.6183 | 0.6602 | 0.6502 | 0.6833 | 0.5803 | 0.6347 | 0.5797 |
| | S (Ours) | 0.7168 | 0.6653 | 0.6859 | 0.6481 | 0.6855 | 0.6797 | 0.7156 | **0.6003** | **0.6443** | **0.5965** |

Table 2: Performance (Test Accuracies as Normalized Pearson Correlation) of various Task Optimized vision models with Linear Ridge (R), Spatial-Feature Factorized Linear (F), Semantic Spatial Transformer (S) and Gaussian2D (G) readouts

| Readout | V1v | V1d | V2v | V2d | V3v | V3d | V4 | Ventral | Dorsal | Lateral |
|---|---|---|---|---|---|---|---|---|---|---|
| R | 0.7746 | 0.7427 | 0.7299 | 0.6906 | 0.6867 | 0.6865 | 0.6551 | 0.4657 | 0.4824 | 0.4372 |
| G | 0.7306 | 0.6744 | 0.6746 | 0.6253 | 0.6326 | 0.6104 | 0.6297 | 0.4784 | 0.4728 | 0.4545 |
| F | 0.83154 | 0.7926 | 0.7795 | 0.7419 | 0.7268 | 0.7323 | 0.7085 | 0.4847 | 0.4831 | 0.4504 |
| S (Ours) | **0.8698** | **0.8340** | **0.8302** | **0.7919** | **0.7808** | **0.7913** | **0.7729** | **0.5796** | **0.6089** | **0.5638** |

Table 3: Performance (Test Accuracies as Normalized Pearson Correlation) of Response Optimized vision models with Linear Ridge (R), Spatial-Feature Factorized Linear (F), Semantic Spatial Transformer (S) and Gaussian2D (G) readouts

| Model Details | | Visual Cortex Region | | | | | | | | | |
|---|---|---|---|---|---|---|---|---|---|---|---|
| LLM | Readout | V1v | V1d | V2v | V2d | V3v | V3d | V4 | Ventral | Dorsal | Lateral |
| Single Caption Models | | | | | | | | | | | |
| C | R | **0.3974** | **0.3779** | **0.3809** | **0.3702** | **0.4093** | **0.4119** | **0.4882** | 0.5661 | 0.6243 | 0.5920 |
| M | R | 0.3931 | 0.3738 | 0.3738 | 0.3687 | 0.4031 | 0.4077 | 0.4873 | **0.5672** | **0.6269** | **0.6126** |
| Dense Caption Models | | | | | | | | | | | |
| C | R | 0.6597 | 0.6154 | 0.6551 | 0.5953 | 0.6371 | 0.6322 | 0.6621 | 0.5807 | 0.6201 | 0.5761 |
| | G | 0.6783 | 0.6277 | 0.6682 | 0.6207 | 0.6644 | 0.6531 | 0.6905 | 0.5980 | 0.6491 | 0.5943 |
| | F | 0.6919 | 0.6329 | 0.6721 | 0.6183 | 0.6603 | 0.6572 | 0.6927 | 0.5915 | 0.6365 | 0.5781 |
| | S (Ours) | **0.7196** | 0.6590 | **0.6903** | 0.6457 | **0.6897** | 0.6774 | **0.7167** | 0.5953 | **0.6562** | **0.6001** |
| M | R | 0.6557 | 0.5941 | 0.6325 | 0.5732 | 0.6162 | 0.6207 | 0.6493 | 0.5679 | 0.5831 | 0.5502 |
| | G | 0.6840 | 0.6261 | 0.6659 | 0.6207 | 0.6583 | 0.6519 | 0.6928 | 0.5934 | 0.6441 | 0.5894 |
| | F | 0.6889 | 0.6339 | 0.6679 | 0.6183 | 0.6602 | 0.6502 | 0.6833 | 0.5803 | 0.6347 | 0.5797 |
| | S (Ours) | 0.7168 | **0.6653** | 0.6859 | **0.6481** | 0.6855 | **0.6797** | 0.7156 | **0.6003** | 0.6443 | 0.5965 |

Table 4: Performance (Test Accuracies as Normalized Pearson Correlation) of language models with Linear Ridge (R), Spatial-Feature Factorized Linear (F), Semantic Spatial Transformer (S) and Gaussian2D (G) readouts

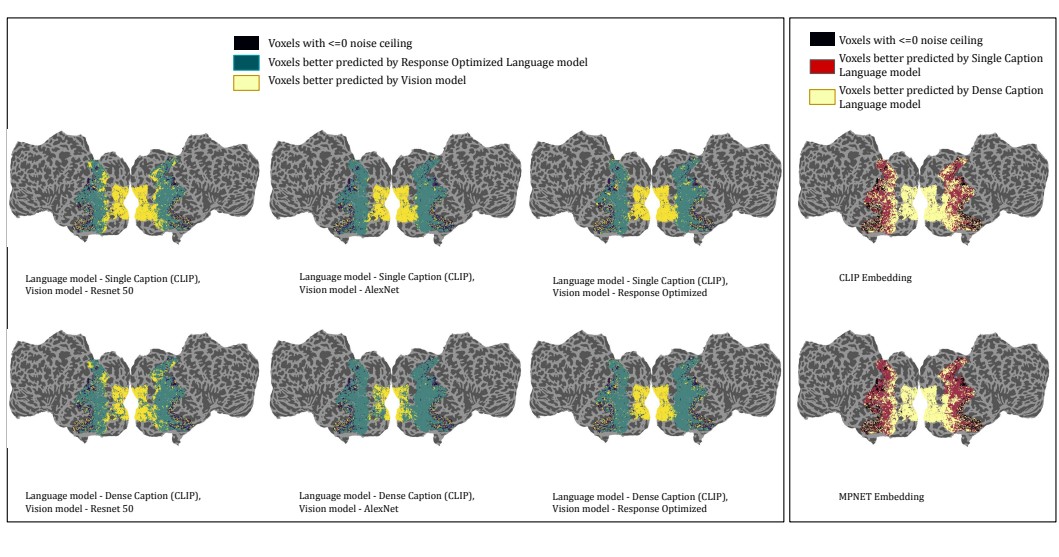

Figure 5: A - Brain Visualizations showing voxels that are better predicted by vision and language models, all using Ridge Linear readouts, B - Brain Visualizations showing voxels that are better predicted by single caption and dense caption language models, all using Ridge Linear readouts

## A.1 NATURAL SCENES DATASET

A detailed description of the Natural Scenes Dataset (NSD; http://naturalscenesdataset.org) is provided elsewhere (Allen et al., 2022). The NSD dataset contains measurements of fMRI responses from 8 participants who each viewed 9,000–10,000 distinct color natural scenes (22,000–30,000 trials) over the course of 30–40 scan sessions. Scanning was conducted at 7T using whole-brain gradient-echo EPI at 1.8-mm resolution and 1.6-s repetition time. Images were taken from the Microsoft Common Objects in Context (COCO) database (Lin et al., 2014), square cropped, and presented at a size of 8.4° x 8.4°. A special set of 1,000 images were shared across subjects; the remaining images were mutually exclusive across subjects. Images were presented for 3 s with 1-s gaps in between images. Subjects fixated centrally and performed a long-term continuous recognition task on the images. The fMRI data were pre-processed by performing one temporal interpolation (to correct for slice time differences) and one spatial interpolation (to correct for head motion). A general linear model was

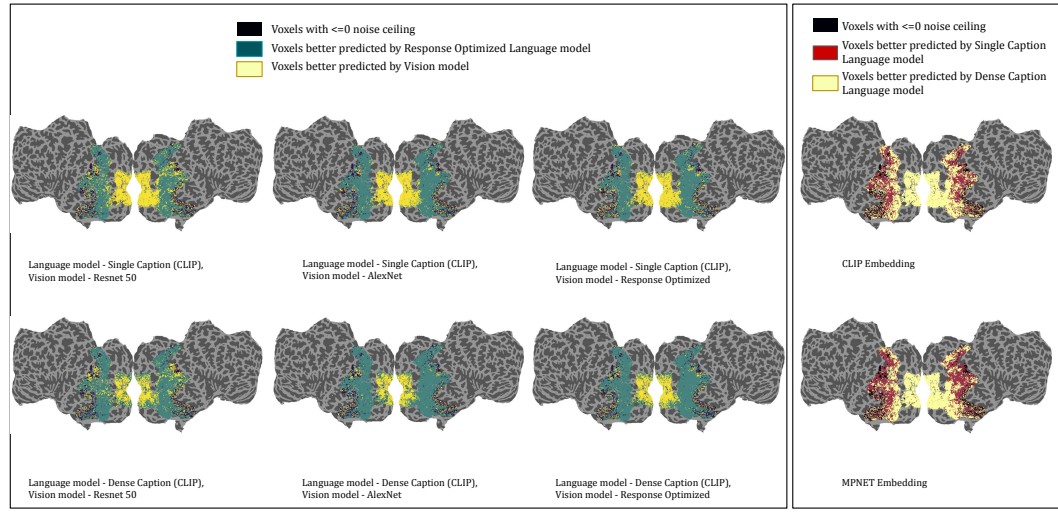

Figure 6: A - Brain Visualizations showing voxels that are better predicted by vision and language models, all using Gaussian2D readouts, B - Brain Visualizations showing voxels that are better predicted by single caption and dense caption language models, all using gaussian2D readouts

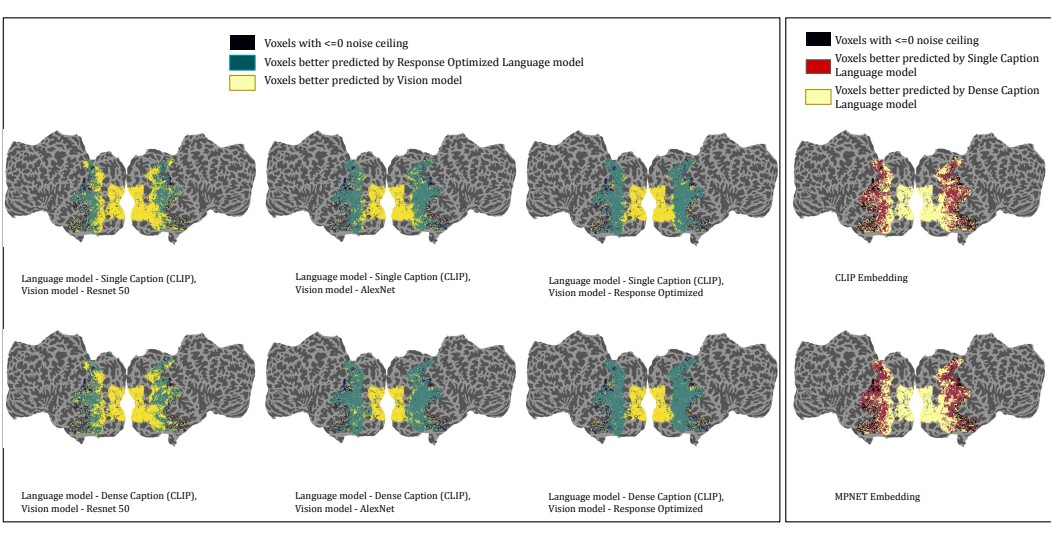

Figure 7: A - Brain Visualizations showing voxels that are better predicted by Vision and language models, all using Spatial-Feature Factorized Linear readouts, B - Brain Visualizations showing voxels that are better predicted by single caption and dense caption language models, all using Spatial-Feature Factorized Linear readouts

then used to estimate single-trial beta weights. Cortical surface reconstructions were generated using FreeSurfer, and both volume- and surface-based versions of the beta weights were created. In this study, we analyze manually defined regions of interest (ROIs) across both early and higher-level visual cortical areas. For early visual areas, we focus on ROIs delineated based on the results of the population receptive field (pRF) experiment - V1v, V1d, V2v, V2d, V3v, V3d, and hV4. For higher

level visual cortex regions, we target the ventral, dorsal, and lateral streams, as defined by the streams atlas.

**Noise Ceiling Estimation in NSD** - Noise ceiling for every voxel represents the performance of the "true" model underlying the generation of the responses (the best achievable accuracy) given the noise in the fMRI measurements. They were computed using the standard procedure followed in (**?**) by considering the variability in voxel responses across repeat scans. The dataset contains 3 different responses to each stimulus image for every voxel. In the estimation framework, the variance of the responses, $\sigma^2_{\text{response}}$, are split into two components, the measurement noise $\sigma^2_{\text{noise}}$ and the variability between images of the noise free responses $\sigma^2_{\text{signal}}$.

$$\hat{\sigma}^2_{\text{response}} = \hat{\sigma}^2_{\text{signal}} + \hat{\sigma}^2_{\text{noise}}$$

An estimate of the variability of the noise is given as $\hat{\sigma}^2_{\text{noise}} = \frac{1}{n} \sum_{i=1}^{n} \text{Var}(\beta_i)$, where i denotes the image (among $n$ images) and $\text{Var}(\beta_i)$ denotes the variance of the response across repetitions of the same image. An estimate of the variability of the noise free signal is then given as,

$$\hat{\sigma}^2_{\text{signal}} = \hat{\sigma}^2_{\text{response}} - \hat{\sigma}^2_{\text{noise}}$$

Since the measured responses were z-scored, $\hat{\sigma}^2_{\text{response}} = 1$ and $\hat{\sigma}^2_{\text{signal}} = 1 - \hat{\sigma}^2_{\text{noise}}$. The noise ceiling (n.c.) expressed in correlation units is thus given as $n.c. = \sqrt{\frac{\hat{\sigma}^2_{\text{signal}}}{\hat{\sigma}^2_{\text{signal}} + \hat{\sigma}^2_{\text{noise}}}}$. The models were evaluated in terms of their ability to explain the average response across 3 trials (i.e., repetitions) of the stimulus. To account for this trial averaging, the noise ceiling is expressed as $n.c. = \sqrt{\frac{\hat{\sigma}^2_{\text{signal}}}{\hat{\sigma}^2_{\text{signal}} + \hat{\sigma}^2_{\text{noise}}/3}}$. We computed noise ceiling using this formulation for every voxel in each subject and expressed the noise-normalized prediction accuracy (R) as a fraction of this noise ceiling.

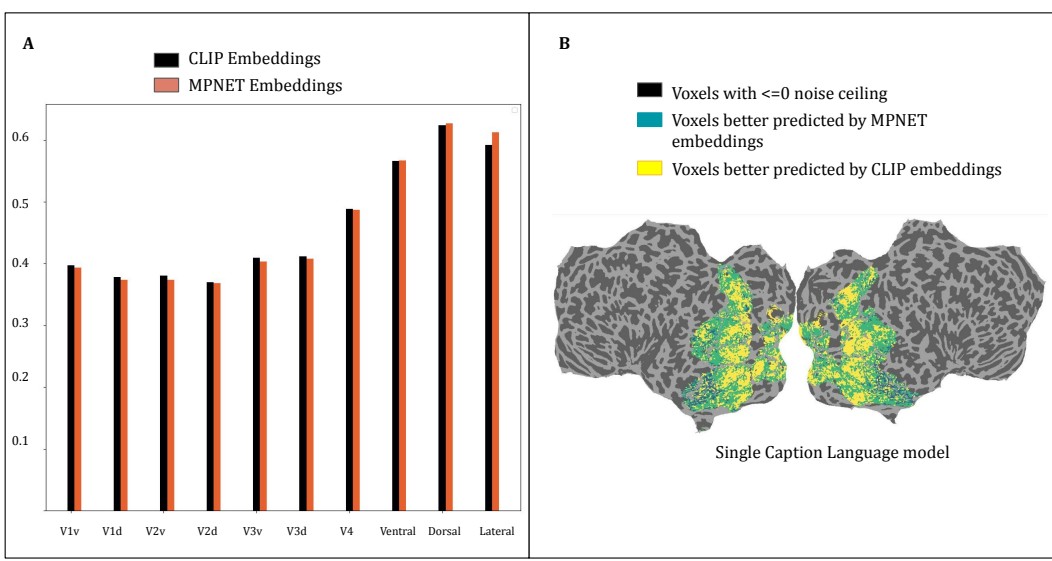

Figure 8: Comparison of Unimodal and Multimodal embeddings in Language models, A - Test Accuracy (Normalized Pearson Correlation) on held out dataset using Single Caption Language encoders with CLIP and MPNET embeddings, B - Brain Visualization showing regions better predicted by each encoder in Single Caption Language models

## A.2 UNIMODAL VERSUS MULTIMODAL EMBEDDING IN LANGUAGE MODELS

As outlined in the previous section, the higher-level regions of the ventral, dorsal, and lateral visual streams exhibit heightened sensitivity to broad semantic information that captures the overall meaning of a scene, as opposed to specific visual details or a combination of visual and spatial features. These

regions are best modeled by single-caption language models. To investigate this further, we examine the performance of models using unimodal encoders like MPNET, which are trained exclusively on language, and multimodal encoders like CLIP, trained on both language and visual data. In the higher regions of the ventral, dorsal, and lateral streams, models using MPNET encoders slightly outperform those with CLIP encoders by 0.5%. This marginal advantage in the higher regions may be attributed to MPNET's optimization for capturing rich semantic nuances from text, aligning well with the language-sensitive nature of these brain regions. On the other hand, in the lower visual regions, where responses are more strongly driven by visual inputs, CLIP encoders hold a small advantage of 1% over MPNET, likely due to their integration of visual knowledge. However, this trend does not hold in dense caption language models, where the performance of both encoders is comparable.

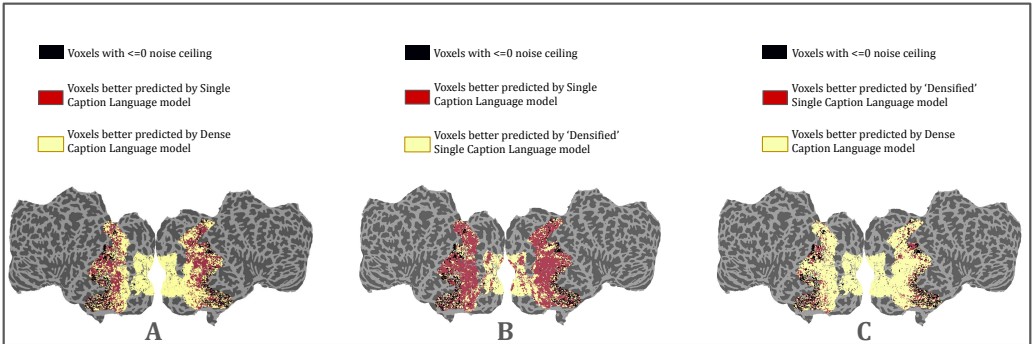

Figure 9: A - Comparison of Single Caption Language models with Dense Caption Language models, B - Comparison of Single Caption Language models with 'Densified' Single Caption Language model, C - Comparison of 'Densified' Single Caption Language model with Dense Caption Language model

### A.3 THE NECESSITY OF SPATIAL SUBDIVISION IN DENSE CAPTIONING FOR EFFECTIVE VISUAL CORTEX MODELING

We further investigated whether the observed differences between dense and global captioning are due to (a) the spatial subdivision of the image (Hypothesis 1) or the increased semantic detail in dense captions (Hypothesis 2). The original idea behind using dense captions was to provide spatial information in addition to semantic information in the form of captions, and subdividing the image into equal sized grids and getting captions for each grid was one of the easiest and most intuitive ways to do that.

We further tried generating more comprehensive single captions of the image using existing LLMs, however none of them were able to provide more information than those already present in the original MS-COCO dataset. In an attempt to densify the single captions, we thus adopted a different approach: for each image, we took the embeddings of dense captions generated for individual grid locations and averaged these embeddings to produce a single "aggregate dense caption" embedding.

On comparing single caption stimuli with 'densified' single caption stimuli (as opposed to the dense caption approach discussed in the paper) (Figure 9), we saw a similar trend where the higher regions of the visual cortex were better modeled by single caption stimuli. However, the transition in sensitivity from dense to single caption in the middle regions of the ventral, dorsal and lateral stream that is so clearly pronounced when using dense captions is missing when using the above 'densified' single captions. Further comparing 'densified' single captions to dense captions (as proposed in the paper), we saw that the dense captions modeled the overall visual cortex better. Hence, we do feel that adding spatial information to the dense caption is necessary for building more accurate models, be it by sub-dividing the image into grids or via any other way.

| Readout | V1v | V1d | V2v | V2d | V3v | V3d | V4 | Ventral | Dorsal | Lateral |
|---------|------|------|------|------|------|------|------|---------|--------|---------|
| F | 0.83154 | 0.7926 | 0.7795 | 0.7419 | 0.7268 | 0.7323 | 0.7085 | 0.4847 | 0.4831 | 0.4504 |
| F + 1 | 0.8596 | 0.8217 | 0.8179 | 0.7769 | 0.7705 | 0.7719 | 0.7659 | 0.5638 | 0.5962 | 0.5371 |
| F + 2 | 0.8750 | 0.8409 | 0.8310 | 0.7948 | 0.7814 | 0.7958 | 0.7782 | 0.5865 | 0.6156 | 0.5641 |

Table 5: Performance (Test Accuracies as Normalized Pearson Correlation) of Spatial-Feature Factorized Linear Readout (F) with individual affine transformations applied to encoder feature maps (1) and spatial masks (2) separately, all with Response Optimized Vision models.

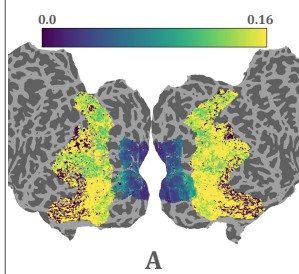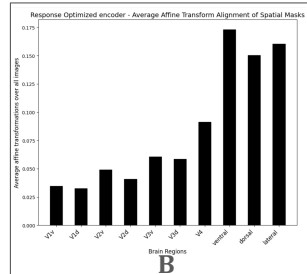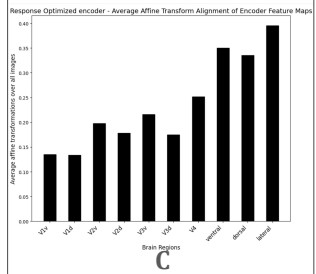

Figure 10: A - Average spatial shifts of voxel spatial masks across all images, B - Mean spatial shifts for each brain region, comparing spatial masks across all images, C - Mean spatial shifts for each brain region, comparing feature maps across all images.

## A.4 Analyzing Spatial Modulation of Receptive Fields in Visual Cortex: Insights from STN Readouts

In an additional experiment focused on interpreting the STN readouts, we calculated the distance between the affine parameters corresponding to the spatial maps of each voxel for every image, relative to the mean affine parameters across all images (Figure 10). The L2 norm of this vector was computed for each voxel. Across all encoders, we observed that stimulus-dependent spatial shifts of the receptive field increase from lower to higher visual regions. A similar trend emerged when calculating the average spatial shifts for each channel of the feature map across images for different regions. This trend further supports the idea that higher levels of the visual cortex benefit more from learned geometric invariances and exhibit greater spatial modulation of their visual receptive fields compared to lower visual cortex regions. This modulation includes phenomena such as receptive field expansion, contraction, or shifts in response to different stimuli.

| Readout | V1v | V1d | V2v | V2d | V3v | V3d | V4 | Ventral | Dorsal | Lateral |
|---------|------|------|------|------|------|------|------|---------|--------|---------|
| F (28*28) | 0.8315 | 0.7926 | 0.7795 | 0.7419 | 0.7268 | 0.7323 | 0.7085 | 0.4847 | 0.4831 | 0.4504 |
| S (28*28) | 0.8698 | 0.8340 | 0.8302 | 0.7919 | 0.7808 | 0.7913 | 0.7729 | 0.5796 | 0.6089 | 0.5638 |
| S (4*4) | 0.8432 | 0.8089 | 0.8056 | 0.7690 | 0.7672 | 0.7743 | 0.7425 | 0.5734 | 0.5986 | 0.5513 |
| S (4*4) | 0.7783 | 0.7328 | 0.7374 | 0.6991 | 0.7061 | 0.7043 | 0.7102 | 0.5699 | 0.6002 | 0.5532 |

Table 6: Performance (Analysis of the effect of channel size on the improvement introduced by Semantic Spatial Transformer Readout (S) over Spatial-Linear Factorized Readouts (F), all with Response Optimized Vision models

## A.5 Dependency of Semantic Spatial Transformer Readout on Channel Size

We acknowledge the importance of ensuring that the readout does not skew conclusions about neural representations. The larger improvements for vision models stem from their feature representations having greater spatial dimensions than language models, allowing the SST to better leverage the rich spatial information available in vision models. To mitigate this, we can normalize spatial

dimensions across models to ensure uniform treatment. Empirically we show that if we reduce the spatial dimensions of the vision encoder to match those of the language encoder, that does drop the prediction performance and relative gains (Table 6).

The overall trend where higher cortical areas are better modeled by language input and lower cortical areas by visual input is consistently observed across all readouts (Figure. 4, 5, 6, 7). However, the margin distinguishing the effectiveness of the models varies slightly. Notably, as we progress from less biologically intuitive readouts to more biologically plausible ones (linear regression, Gaussian 2D, Spatial-Feature Factorized Linear Readout, and finally, the Semantic Spatial Transformer Readout), these trends become increasingly well-defined. Given that the Semantic Spatial Transformer Readout most accurately and consistently models neural responses, we rely on it to delineate regions of the visual cortex sensitive to varying kinds of stimulus information.

| Encoder Type | V1v | V1d | V2v | V2d | V3v | V3d | V4 | Ventral | Dorsal | Lateral |
|---|---|---|---|---|---|---|---|---|---|---|
| A | 0.8507 | 0.8083 | 0.8057 | 0.7603 | 0.7612 | 0.7763 | 0.7674 | 0.6105 | 0.6606 | 0.5823 |
| B | 0.7579 | 0.7034 | 0.7021 | 0.6646 | 0.6861 | 0.6712 | 0.6991 | 0.5546 | 0.5814 | 0.5470 |
| C | 0.8543 | 0.8144 | 0.8084 | 0.7693 | 0.7680 | 0.7772 | 0.7793 | 0.6077 | 0.6764 | 0.5987 |
| D | 0.8147 | 0.7654 | 0.7621 | 0.7163 | 0.7089 | 0.6898 | 0.7114 | 0.5648 | 0.5841 | 0.5469 |
| E | 0.8698 | 0.8340 | 0.8302 | 0.7919 | 0.7808 | 0.7913 | 0.7729 | 0.5796 | 0.6089 | 0.5638 |

Table 7: Performance (Analysis of different architectures for Response and Task Optimized models (A - Task Optimized Resnet 50 (pretrained with ImageNet), B - Response Optimized Resnet 50, C - Task Optimized Mask-RCNN (pretrained with MS-COCO), D - Response Optimized Mask-RCNN, E - Response Optimized E2cnn (proposed), all with Semantic-Spatial Transformer Readouts.

### A.6 COMPARING ARCHITECTURAL APPROACHES FOR TASK AND RESPONSE OPTIMIZED MODELS

Our study carefully controlled several factors to compare task-optimized and response-optimized neural network models for predicting brain responses. Specifically, we held constant both the stimulus set and readout layer, varying only the encoder architecture across models. The rationale for employing different architectures in our study was to leverage state-of-the-art approaches tailored to distinct modeling paradigms. A direct comparison between task-optimized and response-optimized models is inherently challenging due to differences in the available training stimulus sets. Specifically, the stimulus set for training response-optimized models is substantially smaller—approximately 0.03 times the size of the datasets used for task optimization. Incorporating structural biases into response-optimized models (e.g., rotation equivariance) enables them to learn effectively from smaller datasets. This advantage of rotation-equivariant architectures in neural encoding contexts has been demonstrated in prior studies Khosla and Wehbe (2022) and is a critical factor when designing models that align with the constraints of neural data.

While head-on comparisons using identical architectures for task and neural response optimization could provide valuable insights into the specific contributions of these factors , the primary objective of our study was not to isolate these factors. Instead, we aimed to identify the most predictive models for voxel responses across distinct regions of the visual system. Our findings reveal the current best-performing models for this goal, emphasizing practical predictive utility rather than dissecting the contributions of task versus response optimization in isolation.

We conducted further experiments using - a ResNet-50 encoder trained from scratch exclusively on the NSD dataset, a Mask-RCNN encoder trained from scratch on the NSD dataset, a pretrained Mask-RCNN encoder finetuned on the NSD dataset, and compared it with the proposed task and response optimized encoders in the paper all paired with a Semantic Spatial Transformer readout (Table 7). We did this to analyze if the same architecture for response- and task-optimized vision models could provide valuable insights. Unlike the task-optimized ResNet-50, which is trained for object classification on ImageNet, the ResNet-50 trained from scratch on neural responses struggled to match the performance of the proposed response-optimized e2cnn model. The task optimized Mask-RCNN model is pretrained on the MS-COCO dataset which is a superset of the images in the NSD dataset. Although both the task optimized performance show a very similar performance, we once again see a similar trend here with the Mask-RCNN encoder trained from scratch on the NSD dataset, where it struggled to reach the performance of the response optimised e2cnn model.

This comparison underscores the role of network architecture and the significance of incorporating relevant structural biases into networks when optimizing them on response prediction with limited data (atleast in comparison to large-scale vision datasets).

In summary, task optimized models consist of a pretrained CNN core that has been pretrained extensively on large datasets such as ImageNet for very specific object oriented tasks. Only the final layer of these models are finetuned to predict brain responses. While it would be insightful to explore how different task optimized models pretrained on various tasks would perform (as already mentioned in discussion), such a comparison is beyond the scope of this study.Further, past research Conwell et al. (2022a) have shown that varying architectures (convolutional such as Mask-RCNN vs non-convolutional such as transformers) pretrained on different training objectives have minimal impact on brain predictivity. However, task-optimized models tend to be highly biased toward their specific training tasks, which limits their capacity for novel insights into brain function. This limitation led to the development of response-optimized models, which are trained from scratch on neural data with approximately 3% of the dataset size used for task-optimized models. The architecture needs to be very biologically intuitive to model the brain as closely as possible, especially since the training diet is so small. Merely training any architecture (e.g., AlexNet, ResNet-50, or transformers) on neural data does not make it inherently suited for response optimization. Thus, the difference between task- and response-optimized models goes beyond whether one is pretrained or not. Instead, it's about selecting the most effective pretrained model (with optimized architecture and dataset) for task-optimized applications versus the most suitable architecture for response-optimized models that aligns with neural data constraints.

