# OpenReview forum: "Modeling the Human Visual System: Comparative Insights from Response-Optimized and Task-Optimized Vision Models, Language Models, and different Readout Mechanisms"
_ICLR.cc/2025/Conference — Submitted to ICLR 2025_

### Official Review · Reviewer_rgy2 · 2024-10-24

**Soundness:** 3
**Presentation:** 3
**Contribution:** 3
**Rating:** 6
**Confidence:** 3

**Summary:**

The authors aim to evaluate the extent to which LLMs (based on single or dense image captions) predict activity in high-level visual cortex relative to ImageNet-pretrained vision models or neural-response optimized vision models. They introduce a novel readout method that shows higher performance in predicting neural responses relative to linear regression and two other readout methods. They find three distinct regions of visual cortex that are better predicted by vision models, LLMs based on dense captioned images, or LLMs for single image captions.

**Strengths:**

Literature review is comprehensive, and overall, the paper is clearly written.

The paper is not highly original building on prior readout methods, and recent work conducting large-scale benchmarking of AI models against the brain. However, the addition of dense image captions to extract representations from the language models is a nice contribution to the literature on LM alignment with visual cortex. I have some reservations mentioned below that are impacting my score, but if addressed, I think the paper may constitute a meaningful contribution to the literature.

**Weaknesses:**

The paper needs to better justify why a different readout method is necessary. The authors state that the predominant readout method is linear ridge regression, which has high computational and memory demands, but representational similarity analysis (RSA) is nearly as commonly used in the human literature and is less computationally intensive (Kriegeskorte et al, 2008). More importantly, however, the reason that the NeuroAI field tends to rely on linear regression as a readout is based on the logic that we are interested in evaluating the similarity of the representations up to a linear transformation (in representation space) without introducing non-linearities in the readout method. The authors should provide better justification for why a novel readout method is needed within that framework.

The Semantic Spatial Transformer has greater improvement relative to Ridge regression for the vision model than the language model (Figure 2), and vision models are found to better predict more voxels in high-level visual cortex using the Semantic Spatial Transformer readout (Figure 4) than when using Ridge regression (Figure 5). To me, it is a problem that the readout method does not provide uniform improvements across models. This suggests to me that the readout method is introducing a bias in the conclusions. However, I welcome rebuttal on why this logic is faulty.

Figure 4D shows three regions that respond more to vision model, single captions language model, or dense caption language model, but this is binarizing the difference between each pair of models. However, the claims of these three distinct regions would be strengthened by showing that high-level visual voxels, for example, have additional explained variance by the single caption language model after accounting for the variance explained by vision and dense caption models.

**Questions:**

Why did the authors only use 4 of the 8 participants from NSD?

Figure 1A is confusing. I don’t follow how each of the different readout methods are shown here. Better labels would be very helpful.

---

> ### Author Response · Authors · 2024-11-22
> **Official Response to Reviewer rgy2**
>
> We would like to thank the reviewers for taking the time out to review our paper, and for raising a lot of important questions. First, we would like to address the questions raised -
>
> **Why did the authors only use 4 of the 8 participants from NSD?**
>
> The fmri responses were similar across the various subjects, and it is common practice to train models on these 4 datasets only [1], [2], [3]. The data from the held out subjects is usually used for fine tuning or zero shot tasks as seen in [1]. Also, these were the only subjects that completed all 40 NSD sessions.
>
> **Figure 1A is confusing. I don’t follow how each of the different readout methods are shown here. Better labels would be very helpful.**
>
> We apologize for the confusion, and have attempted to make this figure more clear in the next revision of the paper. It will be updated very soon.
>
> **References**
>
> 1. Khosla, Meenakshi, and Leila Wehbe. "High-level visual areas act like domain-general filters with strong selectivity and functional specialization." bioRxiv (2022): 2022-03.
> 2. Joo, Jaehoon, Taejin Jeong, and Seongjae Hwang. "Brain-Streams: fMRI-to-Image Reconstruction with Multi-modal Guidance." arXiv preprint arXiv:2409.12099 (2024).
> 3. Liu, Yulong, et al. "See Through Their Minds: Learning Transferable Neural Representation from Cross-Subject fMRI." arXiv preprint arXiv:2403.06361 (2024).
> 4. Doerig, Adrien, et al. "Semantic scene descriptions as an objective of human vision." arXiv preprint arXiv:2209.11737 10 (2022).
>
> *Rebuttals to Weaknesses continued in the next response.*

---

> ### Author Response · Authors · 2024-11-22
> **Continued Official Response to Reviewer rgy2 (Weakness 1)**
>
> **Weakness 1 -** *The paper needs to better justify why a different readout method is necessary. The authors state that the predominant readout method is linear ridge regression, which has high computational and memory demands, but representational similarity analysis (RSA) is nearly as commonly used in the human literature and is less computationally intensive (Kriegeskorte et al, 2008). More importantly, however, the reason that the NeuroAI field tends to rely on linear regression as a readout is based on the logic that we are interested in evaluating the similarity of the representations up to a linear transformation (in representation space) without introducing non-linearities in the readout method. The authors should provide better justification for why a novel readout method is needed within that framework.*
>
> First, we would like to clarify the distinction between readout models and similarity metrics, as they serve fundamentally different purposes in computational neuroscience. In summary, RSA is a post hoc analysis tool for measuring representational similarity, whereas readout models are predictive frameworks that learn mappings between neural network representations and fMRI responses. Thus, the two methodologies are not interchangeable and address distinct aspects of computational modeling in neuroscience.
>
> - **Readout models** are designed to map the feature representations from a neural network encoder to voxel-wise fMRI responses in specific brain regions. These models perform a predictive function by directly aligning network features with measured neural responses. Predictive modeling can be a very useful goal in many contexts. For e.g., accurate and robust neural network models are essential for in-silico experiments [2] to explore hypotheses that would be challenging or impractical to test in vivo, inform experimental design, as well as for precise neural population control. In this way, accuracy provides a foundational utility for practical and theoretical advancements.
> - The most popular readout method in the human visual neuroscience literature is a linear readout, wherein the features are flattened/vectorized and linearly combined to predict responses (often with an additional l2 regularization on the feature weights ala ridge regression). However, the separability of what is computed from where it is computed (spatial receptive field) which is a fundamental visual neuroscience finding is not reflected in the ridge regression readouts that operated on flattened feature representations. More constrained domain-specific readout approaches, such as Spatial X Feature Factorized or Gaussian 2D readouts (which were previously proposed in the context of mouse visual cortex modeling) are also linear mapping models but they separate a neuron/voxel’s spatial receptive field (what spatial locations in the encoder output does a voxel prefer) from the voxel’s featural preferences (what channels capture its featural selectivity). In this way, they are less expressive readouts than the ridge regression readout but have strong biological grounding. Empirically, we show that the  Spatial X Feature Factorized consistently outperforms the much more expressive ridge regression readout. Now, coming to our proposed readout, the Semantic Spatial Transformer (SST readout) addresses a significant limitation of other readouts: other readouts assume that the spatial receptive field remains constant irrespective of the stimuli. The SST readout on the other hand enables dynamic adjustments of spatial receptive fields depending on stimulus context. We expand on this below. More details about these methods can be found in our paper in section 2.2.
> - On the other hand, similarity assessment tools such as **Representational Similarity Analysis (RSA)** serve a different role. RSA is a powerful tool used to evaluate how similar two sets of high-dimensional representations are, by comparing their pairwise representational geometries. For example, RSA computes a similarity matrix (e.g., correlation or cosine similarity) for the true and predicted voxel representations, allowing researchers to assess whether the underlying structure of the representations aligns. However, RSA does not provide a direct mapping from features to voxel responses, and lacks the predictive functionality required of a readout model.
>
> While it is true that RSA is computationally less intensive and widely used in the literature (e.g., Kriegeskorte et al., 2008), its purpose is fundamentally different. It is not a substitute for a readout model because it does not establish a voxel-wise predictive mapping. Instead, it is complementary to readout models, providing a means to evaluate the alignment of representational structures after the mapping is established.
>
> *Response Continued in next comment.*

---

> ### Author Response · Authors · 2024-11-22
> **Continued Official Response to Reviewer rgy2 (Weakness 1 Continued)**
>
> We acknowledge that the biological motivations for the STN were perhaps not sufficiently emphasized in the initial version of the paper. In the next revised version, we would add the below details.
>
> We completely agree with the reviewer that one of the main reasons behind the widespread use of linear regression as a readout model is because it does not introduce non-linearities in the encoder feature maps, as we want to find out how similar the encoder feature maps are to the brain responses without any external finetuning. However, as already mentioned in the paper, one of the major disadvantages of this readout is its computational complexity. The other readouts (including our proposed readout) also linearly map features to brain responses but with additional constraints on the structure of the weight space in line with biological findings.
>
> We take the Spatial-Feature Factorized Linear Readout a step further with our proposed STN readout. We first emphasize that the STN does not involve unnecessarily complex transformations. Instead, it only spatially modulates the feature maps and spatial masks, allowing for dynamic and stimulus-dependent adjustments. Once again, no non-linearities are introduced here. The STN allows voxel-specific spatial modulation of receptive fields (RFs), inspired by studies demonstrating the dynamic adaptability of RFs to stimulus properties:
> * RF sizes can expand or contract based on contrast (Sceniak et al., Nature Neuroscience, 1999).
> * RFs can also shift or reshape in response to contextual or attentional influences (Womelsdorf et al., Nature Neuroscience, 2006).
>
> Unlike fixed spatial masks used in linear readouts (e.g., Spatial-Feature Factorized Linear or Gaussian 2D readouts), the STN employs affine transformations to capture stimulus-dependent spatial changes such as translation, scaling, rotation, and shearing. This flexibility enables more accurate modeling of neural responses that exhibit dynamic RF properties.
> The STN also spatially transforms feature maps and spatial weights at the channel level. Each channel typically encodes distinct attributes (e.g., edges, textures, shapes), and the ability to apply channel-specific transformations allows the STN to adapt to their unique geometric properties. For instance, one channel may require scaling to emphasize fine-grained details, while another might need rotation for orientation invariance. This flexibility is particularly advantageous for neural response modeling. Unlike object classification tasks, where we can employ augmentations tailored to known invariances (e.g. rotating an image won’t change the category label) to boost predictive accuracy, the geometric invariances of voxel responses are unknown. STN enables the network to learn these invariances directly from the data, providing a crucial advantage in predicting voxel responses across diverse visual regions.
>
> *Response Continued in next comment.*

---

> ### Author Response · Authors · 2024-11-22
> **Continued Official Response to Reviewer rgy2 (Weakness 2)**
>
> **Weakness 2 -** *The Semantic Spatial Transformer has greater improvement relative to Ridge regression for the vision model than the language model (Figure 2), and vision models are found to better predict more voxels in high-level visual cortex using the Semantic Spatial Transformer readout (Figure 4) than when using Ridge regression (Figure 5). To me, it is a problem that the readout method does not provide uniform improvements across models. This suggests to me that the readout method is introducing a bias in the conclusions. However, I welcome rebuttal on why this logic is faulty.*
>
> We appreciate the reviewer’s concern about potential bias introduced by the Semantic Spatial Transformer (SST) readout method. We acknowledge the importance of ensuring that the readout does not skew conclusions about neural representations.
> The larger improvements for vision models stem from their feature representations having greater spatial dimensions than language models, allowing the SST to better leverage the rich spatial information available in vision models. To mitigate this, we can normalize spatial dimensions across models to ensure uniform treatment. Empirically we show below that if we reduce the spatial dimensions of the vision encoder to match those of the language encoder, that does drop the prediction performance and relative gains.
>
> | Feature Map Channel Size | Readout                         | V1v   | V1d   | V2v   | V2d   | V3v   | V3d   | V4    | Ventral | Dorsal | Lateral |
> |--------------------------|---------------------------------|-------|-------|-------|-------|-------|-------|-------|---------|--------|---------|
> | 28×28                    | Spatial Feature Factorized      | 0.8315| 0.7926| 0.7795| 0.7419| 0.7268| 0.7323| 0.7085| 0.4847  | 0.4831 | 0.4504  |
> | 28×28                    | Semantic Spatial Transformer    | 0.8698| 0.8340| 0.8302| 0.7919| 0.7808| 0.7913| 0.7729| 0.5796  | 0.6089 | 0.5638  |
> | 4×4 (A)                  | Semantic Spatial Transformer    | 0.8432| 0.8089| 0.8056| 0.7690| 0.7672| 0.7743| 0.7425| 0.5734  | 0.5986 | 0.5513  |
> | 4×4 (B)                  | Semantic Spatial Transformer    | 0.7783| 0.7328| 0.7374| 0.6991| 0.7061| 0.7043| 0.7102| 0.5699  | 0.6002 | 0.5532  |
>
> **Notes:**
> - **A** - Original 28×28 feature maps are downsampled via bilinear interpolation to 4×4.
> - **B** - Smaller image sizes (32×32) are used instead of (224×224) to get a smaller 4×4 feature map.
>
> The overall trend—where higher cortical areas are better modeled by language input and lower cortical areas by visual input—is consistently observed across all readouts (Fig. 4, 5, 6, 7). However, the margin distinguishing the effectiveness of the models varies slightly. Notably, as we progress from less biologically intuitive readouts to more biologically plausible ones (linear regression, Gaussian 2D, Spatial-Feature Factorized Linear Readout, and finally, the Semantic Spatial Transformer Readout), these trends become increasingly well-defined. Given that the Semantic Spatial Transformer Readout most accurately and consistently models neural responses, we rely on it to delineate regions of the visual cortex sensitive to varying kinds of stimulus information.
>
> Further, we note that this issue is not unique to the SST readouts. Ridge regression itself can favor larger models due to their higher number of basis features, potentially inflating prediction scores without necessarily reflecting greater alignment with the brain, as noted in prior research (Schaeffer et al., 2024). Thus, biases in readouts are a general concern, and the SST readout is not uniquely problematic in this regard.
>
> **Reference:  https://openreview.net/pdf?id=vbtj05J68r**
>
> **Please refer to Supplementary Section A.4 and Figure 10 for additional analysis on the SST readout, in the current revision of the paper.**
>
> *Response Continued in next comment.*

---

> > ### Author Response · Authors · 2024-11-22
> > **Continued Official Response to Reviewer rgy2 (Weakness 3)**
> >
> > **Weakness 3 -** *Figure 4D shows three regions that respond more to vision model, single captions language model, or dense caption language model, but this is binarizing the difference between each pair of models. However, the claims of these three distinct regions would be strengthened by showing that high-level visual voxels, for example, have additional explained variance by the single caption language model after accounting for the variance explained by vision and dense caption models.*
> >
> > Thank you for your insightful suggestion. We agree that computing the shared and unique variance explained by each model would strengthen our claims. However, implementing this would require training linear readouts after concatenating all feature spaces (e.g., using methods such as banded ridge regression). Given the high-dimensional representations, particularly those of dense caption language models, this approach would be computationally prohibitive. While it is an important direction for future work, we opted for the current analysis to balance clarity and computational feasibility.

---

> > > ### Comment · Reviewer_rgy2 · 2024-11-24
> > >
> > > I appreciate the clarification that the authors provide on **Weakness 1**. However, it does not appear that this additional context has yet been incorporated in the text. This context is critical to readers to understand the contribution of the paper and incorporating it would significantly strengthen the paper. I may be mistaken that it has already been incorporated as the authors have not marked their revisions to text, and I having difficulty tracking updates that have been made.
> > >
> > > I also appreciate discussion on **Weakness 2**. However, the response of the authors has—if anything—has increased my concern of the bias introduced by SST. At a minimum, this needs to be more explicitly incorporated into the limitations section of the main text and for Figure 10 and Section A.4 to be referenced in the main text. I also do not find the authors argument that ridge regression is biased to favor higher-dimensional feature spaces satisfying. They reference a highly controversial non-archival workshop paper to support their argument, and higher dimensional feature spaces have not been found to uniformly enhance brain alignment (Conwell et al., 2024; Elmoznino et al., 2024).
> > >
> > > In the current form, I am not able to update my score.
> > >
> > >
> > > Conwell, C., Prince, J. S., Kay, K. N., Alvarez, G. A., & Konkle, T. (2024). A large-scale examination of inductive biases shaping high-level visual representation in brains and machines. *Nature Communications*, 15(1), 9383. https://doi.org/10.1038/s41467-024-53147-y
> > >
> > > Elmoznino, E., & Bonner, M. F. (2024). High-performing neural network models of visual cortex benefit from high latent dimensionality. PLOS Computational Biology, 20(1), e1011792. https://doi.org/10.1371/journal.pcbi.1011792

---

> ### Author Response · Authors · 2024-11-26
> **Updates for Reviewer rgy2**
>
> We appreciate your acknowledgment that the additional context strengthens the paper. We apologize for not marking the revisions earlier and hope this update clarifies our contributions. We have incorporated the following changes based on your specific suggestions -
>
> 1. *Questions* - **Please refer to the latest version of Figure 1 at *page 3* of our latest revision.**
> 2. *Weakness 1* -
>   - The motivation behind the need of Semantic Spatial Transformer Readouts are added in lines 274-287
>   - Added further analysis in Supplementary section A.4 (Table 5, Figure 10), and they have been referenced in the main text at line 287 - Analyzing Spatial Modulation of Receptive Fields in Visual Cortex: Insights from STN Readouts.
> 3. *Weakness 2* -
>   - **We do understand that you still have concerns about Weakness 2, and we shall respond to them in a later comment.** For now, we have added the current analysis in supplementary section A.5 and referenced them in the main text at lines 333-334.
>
> Here are all the changes we have added throughout the paper based on the various suggestions (**Please refer to the latest revision of the paper**)-
>
> 1. Reorganizing Figure 1 based on the comments received by reviewers **gWb8** and **rgY2** -
>    - Added a brief overview of the entire pipeline (Schematic Pipeline of DNN Models)
>    - Separating out the individual components of the pipeline
>    - Replacing the tensor product symbol with reference to actual equation numbers used in the paper
> 2. Reorganized Figures - 2,3,4 by reducing the unused blank space in the cortical flatmaps according to reviewer **gWb8**.
> 3. Updated Figure 2B with discrete color maps as suggested by reviewers **rgY2** and **EMZd**.
> 4. Updated Figure 2A to highlight the advantages of STN readout as suggested by reviewer **EMZd**.
> 5. Better motivated the utility of Semantic Spatial Transformer Readout (**Line 274-287**) based on the suggestions of reviewers **zmHZ**, **gWb8**, **rgY2** and **EMZd**.
>     - Added further analysis in **Supplementary section A.4 (Table 5, Figure 10)**, and they have been referenced in the main text at **line 287** - *Analyzing Spatial Modulation of Receptive Fields in Visual Cortex: Insights from STN Readouts*.
>     - As suggested by reviewer **rgY2**, we added further analysis on the dependency of the Semantic Spatial Transformer Readouts on channel size (and bias introduced by the readouts) in **Supplementary section A.5 (and Table 6)** - *Dependency of Semantic Spatial Transformer Readout on Channel Size*. This has been referenced in the main text in **line 333-334**.
> 6. As suggested by reviewer **zmHZ**, we added further analysis on the utility of dense captions in **Supplementary Section A.3 (and Figure 9)**, and referenced them in the main text at line **215**.
> 7. As suggested by Reviewer **EMZd**, we added further analysis on *‘Comparing Architectural Approaches for Task and Response Optimized models’* in **Supplementary Section A.6 and Table 7**, and referenced them in the main text at line **192**.

---

> > ### Author Response · Authors · 2024-12-02
> > **Second Official Response to Reviewer rgy2**
> >
> > We appreciate the reviewer’s references to Conwell et al. (2024) and Elmoznino et al. (2024), and would like to clarify some nuances in our argument and address the relevant findings from these works.
> > Firstly, Elmoznino et al. (2024) explicitly state that top-performing encoding models of high-level visual cortex tend to exhibit high latent dimensionality. They argue that this high dimensionality facilitates better performance in tasks requiring generalization, such as learning new categories of stimuli. This suggests that higher-dimensional representations might inherently capture richer feature spaces that align more closely with the neural coding in the brain, particularly in higher-level visual regions. Thus, while dimensionality is not a universal determinant of brain alignment, its benefits have been observed under specific contexts, particularly for generalization and representational richness.
> >
> > Secondly, we wish to draw an important distinction between the notion of "high dimensionality" as discussed by Elmoznino et al. and Conwell et al. and the concept of "larger channel dimensions" favoring the SST readout as referenced in our paper. Dimensionality, in the context of the cited works, typically refers to the number of top principal components or latent features retained in the feature space. Larger channel dimensions in our study, however, relate to the degree of spatial pooling or reduction that the input feature maps undergo during encoding. For example, models with smaller spatial dimensions (e.g., 4×4 feature maps) result from significant pooling, which inherently limits the spatial richness available to the SST readout. This is distinct from the latent dimensionality referred to in the cited studies.
> >
> > Lastly, we recognize the reviewer’s concern regarding potential biases in readouts. It is worth emphasizing that our claim about ridge regression is not merely about dimensionality inflating alignment scores but rather about its general tendency to favor models with richer basis functions - whether through dimensionality (more capacity to encode information) or other factors such as spatial detail. While the main claims of Schaeffer et al., 2024 about the limitations of the neural regression methodology in recovering good models of the brain is controversial, the underlying argument that ridge regression has inductive biases still stands. While this effect of dimensionality is not universally problematic, it warrants attention when comparing models with different spatial or feature characteristics, as seen in our experiments.
> >
> > We acknowledge that the differing spatial dimensions of feature maps in different models could create an issue for model comparison ( We have referenced our new analyses A4 in the main text at line 287). But the spatial dimensions of feature maps can be easily standardized. Future studies aiming to compare models on equal footing—particularly to draw conclusions about computational objectives or architectural influences on neural representations—can address this by maintaining consistent spatial dimensions. Finally, the significantly higher predictive accuracy of the SST readout is a major advantage, especially for applications like in-silico experimentation and neural population control, where improved accuracy provides substantial benefits.
> > In conclusion, we believe the evidence supports our claim that larger channel dimensions inherently benefit the SST readout by providing richer spatial details for alignment, a property not directly analogous to latent dimensionality but complementary in understanding model behavior. We hope this clarifies the reviewer’s concerns.

---

> > > ### Comment · Reviewer_rgy2 · 2024-12-02
> > >
> > > Thank you to authors for their additional engagement on this point. I appreciate the clarification and discussion, and as a result, I have adjusted my score to "marginally above acceptance." My concerns about bias have been discussed but would need to be more seriously investigated in the current manuscript for a stronger recommendation.

---

### Official Review · Reviewer_EMZd · 2024-10-31

**Soundness:** 2
**Presentation:** 3
**Contribution:** 2
**Rating:** 5
**Confidence:** 3

**Summary:**

This paper compares task-optimized vision and language models to brain response-optimized networks on the natural scenes fMRI dataset (NSD), and introduce a new readout mechanism for model-brain mapping based on spatial receptive field and semantic content. They find that response-optimized networks provide the best match to early visual regions, while task optimized vision-language models better match high-level visual regions. They also find their readout mechanism using spatial transformers improves model to brain mapping (though only marginally).

The comparison between response and task optimized models is interesting, but overall the results provide only a marginal advance in our understanding and computational models of visual cortex. The spatial transformer network readout is novel, but it is not entirely clear what the value of this contribution is. It provides slightly better performance over other methods, but is much more complex (involves integrating an additional Resnet-50 module to weight the channels of each model) and provides only minimal quantitative gains over much simpler methods.

**Strengths:**

-	Compare response optimized and task optimized models directly
-	Compared many different model-brain mapping functions
-       Present a new metric for model-brain mapping

**Weaknesses:**

-	The models tested varied along many factors making it difficult to draw strong conclusions about the role of response vs. task-optimization or vision vs. language in model’s performance. For response vs. task, these points could have been made more compelling by training the same architecture on both task and neural responses
-	The biggest issue is that the major findings of this paper have been shown previously (also on the same dataset). Prior work with vision and language models (e.g., Doerig et al.) that showed semantic content is more important for high than low-level visual regions, and Khosla & Wehbe which introduced the response optimized model used here and already showed it predicted NSD responses.
-	The utility of the STN was not well motivated. Figure 2 suggests that the different readout mechanisms provide largely similar results with only minor quantitative differences. This slight boost is unsurprising given how much more complex the spatial transformer network is compared to other readouts.

**Questions:**

-	Why are the models in Figure 2b shown on a continuous color bar?
-	Why were only a subset of the NSD subjects used?

---

> ### Author Response · Authors · 2024-11-19
> **Official Response to Reviewer EMZd (Questions and Weakness 1)**
>
> Thank you for taking the time out to review our paper. First, we would like to answer your questions, and then provide our rebuttal for the weaknesses mentioned -
>
> **1. Why are the models in Figure 2b shown on a continuous color bar?**
>
> We apologize for the inconsistency in this diagram, and it is fixed in our latest revised pdf.
>
> **2. Why were only a subset of the NSD subjects used?**
>
> The fmri responses were similar across the various subjects, and it is common practice to train models on these 4 datasets only [1], [2], [3]. The data from the held out subjects is usually used for fine tuning or zero shot tasks as seen in [1]. Also, these were the only subjects that completed all 40 NSD sessions.
>
> Now, regarding the weaknesses mentioned by the reviewer -
>
> **Addressing Weakness 1:** Although our models were tested across many different factors, when comparing response optimized models with task optimized models, we made sure to keep the stimuli and the readout layer constant. The only variable was the encoder architecture.
>
> The primary reason for employing different architectures in our study was to leverage state-of-the-art approaches tailored to distinct modeling paradigms. A direct comparison between task-optimized and response-optimized models is inherently challenging due to differences in the available training stimulus sets. Specifically, the stimulus set for training response-optimized models is substantially smaller—approximately 0.03 times the size of the datasets used for task optimization.
>
> Incorporating structural biases into response-optimized models (e.g., rotation equivariance) enables them to learn effectively from smaller datasets. This advantage of rotation-equivariant architectures in neural encoding contexts has been demonstrated in prior studies [1] and is a critical factor when designing models that align with the constraints of neural data.
>
> While head-on comparisons using identical architectures for task and neural response optimization could provide valuable insights into the specific contributions of these factors , the primary objective of our study was not to isolate these factors. Instead, we aimed to identify the most predictive models for voxel responses across distinct regions of the visual system. Our findings reveal the current best-performing models for this goal, emphasizing practical predictive utility rather than dissecting the contributions of task versus response optimization in isolation. However, we do agree with the reviewer that using the same architecture for response- and task-optimized vision models could provide a valuable comparison, especially to directly assess their relative sample efficiencies. To address this, we conducted additional experiments using a ResNet-50 encoder paired with a semantic spatial transformer readout, trained from scratch exclusively on the NSD dataset, and compared it with the proposed task and response optimized encoders in the paper all paired with a semantic spatial transformer readout. Unlike the task-optimized ResNet-50, which is trained for object classification on ImageNet, the ResNet-50 trained from scratch on neural responses struggled to match the performance of the response-optimized e2cnn model. This comparison underscores the role of network architecture and the significance of incorporating relevant structural biases into networks when optimizing them on response prediction with limited data (atleast in comparison to large-scale vision datasets).
>
> | Encoder Type (all with semantic spatial transformer readout) |   V1v   |   V1d   |   V2v   |   V2d   |   V3v   |   V3d   |    v4    | Ventral | Dorsal | Lateral |
> |--------------------------------------------------------------|---------|---------|---------|---------|---------|---------|---------|---------|--------|---------|
> | Response Optimized resnet50                                  |  0.7579 |  0.7034 |  0.7021 |  0.6646 |  0.6861 |  0.6712 |  0.6991 |  0.5546 | 0.5814 |  0.5470 |
> | Task optimized resnet50 (proposed)                           |  0.8507 |  0.8083 |  0.8057 |  0.7603 |  0.7612 |  0.7763 |  0.7674 |  0.6105 | 0.6606 |  0.5823 |
> | Response Optimized E2cnn (proposed)                          |  0.8698 |  0.8340 |  0.8302 |  0.7919 |  0.7808 |  0.7913 |  0.7729 |  0.5796 | 0.6089 |  0.5638 |
>
>
> Lastly, language models are predominantly based on transformer architectures, which require significantly larger datasets and computational resources and training response-optimized models using transformer architectures for the visual system  is not feasible.
>
> *Response continued in the next comment.*

---

> ### Author Response · Authors · 2024-11-19
> **Continued Official Response to Reviewer EMZd (Weakness 2)**
>
> *Continued from above comment.*
>
> **Addressing Weakness 2:** We appreciate the reviewer's thoughtful comments and acknowledge the importance of the observations made. We would like to highlight the specific contributions that distinguish our work from those mentioned -
>
> - **Comparison of modeling techniques:** While [4] demonstrates that language models can predict brain activity and that semantic information is important for higher visual regions, their conclusions are primarily based on representational similarity analysis (RSA) rather than direct prediction accuracies. In contrast, our study directly compares three dominant modeling approaches—response-optimized models, task-optimized vision models, and large language model (LLM) embeddings—to identify the most predictive models across the entire visual system. Identifying these models holds significant practical utility for applications such as in-silico experimentation, automated interpretability analyses of voxel responses, and neural population control.
>
> - **Region-specific semantic sensitivities:** We build on the findings of [4] by offering a more nuanced analysis of how different regions of the visual cortex respond to semantic information. Specifically, we show that:
>     - Response-optimized vision models outperform LLM embeddings in mid-level vision regions when using single captions, but this gap is mitigated when dense captions are used. This suggests that mid-level regions encode perceptual information aligned with linguistic descriptions (localized semantics). We identify a transition from regions sensitive to primarily perceptual features (not aligned with language) to those sensitive to localized semantics (aligned with language) and finally to those aligned with global semantics (aligned with language) as we move deeper into the cortical hierarchy.
>     - Based on this analysis, we characterize three distinct regions in the visual cortex, each with varying sensitivities to perceptual input, localized semantics, and global semantics. To the best of our knowledge, such analyses have not been conducted before and represent a key contribution of our work.
>
> - **Extending response-optimized model analyses:** We acknowledge [1] for introducing response-optimized models and showing their ability to predict NSD responses. However, their study primarily focuses on four such regions and does not examine the broader visual system. We do not claim the response-optimized modeling technique as part of our contributions. Instead, we use response-optimized models as a competitive baseline to compare against other models and to identify regions (V1-V4), whose featural selectivities are not yet fully emergent in modern task-optimized models.
>
> - **Comparing different readout methods and introducing a novel readout:** One of our major contributions is the systematic comparison of different readout mechanisms for predicting fMRI responses. We show that the choice of readout significantly affects prediction accuracy and demonstrate how novel domain-specific approaches, such as our proposed STN readout, yield substantial improvements. These findings highlight an underexplored avenue for advancing neural response modeling for complex stimuli. To our knowledge, this comparative analysis of readout methods has not been done before. We highlight the significant quantitative advantages conferred by our proposed STN readout and its biological motivation in our next response.
>
> *Response continued in the next comment.*

---

> > ### Author Response · Authors · 2024-11-19
> > **Continued Official Response to Reviewer EMZd (Weakness 3)**
> >
> > *Continued from above comment.*
> >
> > **Addressing Weakness 3:** Here, we respectfully disagree with the reviewer’s assertion that the proposed readout yields only marginal gains in predictive accuracy.
> >
> > **1. Quantitative gains in accuracy:** The improvements provided by the STN are substantial, especially when compared to other modeling variations. For instance:
> > - Varying encoding strategies (response-optimized vs. task-optimized vs. LLM embeddings) yielded a maximal accuracy difference of ~5%.
> >
> > - In contrast, the choice of readout had a much larger impact. The proposed STN readout improved prediction accuracies by 3.76–26.3% over Spatial-Feature Factorized Linear readouts and by 16.87–52.3% over ridge regression (the de facto choice in many studies) when applied to vision models.
> >
> > In the ventral visual stream, for example, when using response-optimized models:
> >    - Ridge regression achieved a prediction accuracy of 0.46.
> >    - Gaussian and Spatial-Feature Linear Factorized readouts each improved this to 0.48.
> >    - The STN readout further boosted this to 0.58, representing a significant jump over the alternatives.
> >
> > We acknowledge that the gains are less pronounced for language models, which we attribute to the smaller spatial size of the feature maps fed into the readouts, as noted in line 329 of the manuscript.
> >
> > **2. Biological motivation behind the STN readout:** We acknowledge that the biological motivations for the STN were perhaps not sufficiently emphasized in the initial version of the paper. We will add the below further clarifications in the next revision. We first emphasize that the STN does not involve unnecessarily complex transformations. Instead, it only spatially modulates the feature maps and spatial masks, allowing for dynamic and stimulus-dependent adjustments.
> >
> > - **Spatial Modulation of Spatial Masks (Spatial Receptive Fields):** The STN allows voxel-specific spatial modulation of receptive fields (RFs), inspired by studies demonstrating the dynamic adaptability of RFs to stimulus properties:
> >
> >    - RF sizes can expand or contract based on contrast (Sceniak et al., Nature Neuroscience, 1999).
> >    - RFs can also shift or reshape in response to contextual or attentional influences (Womelsdorf et al., Nature Neuroscience, 2006).
> >
> > Unlike fixed spatial masks used in linear readouts (e.g., Spatial-Feature Factorized Linear or Gaussian 2D readouts), the STN employs affine transformations to capture stimulus-dependent spatial changes such as translation, scaling, rotation, and shearing. This flexibility enables more accurate modeling of neural responses that exhibit dynamic RF properties.
> >
> > - **Spatial Modulation of Feature Maps:** The STN also spatially transforms feature maps and spatial weights at the channel level. Each channel typically encodes distinct attributes (e.g., edges, textures, shapes), and the ability to apply channel-specific transformations allows the STN to adapt to their unique geometric properties. For instance, one channel may require scaling to emphasize fine-grained details, while another might need rotation for orientation invariance. This flexibility is particularly advantageous for neural response modeling. Unlike object classification tasks, where we can employ augmentations tailored to known invariances (e.g. rotating an image won’t change the category label) to boost predictive accuracy, the geometric invariances of voxel responses are unknown. STN enables the network to learn these invariances directly from the data, providing a crucial advantage in predicting voxel responses across diverse visual regions.
> >
> > | Readout Type (all with response optimized E2cnn encoder) | V1v    | V1d    | V2v    | V2d    | V3v    | V3d    | V4     | Ventral | Dorsal | Lateral |
> > |----------------------------------------------------------|--------|--------|--------|--------|--------|--------|--------|---------|--------|---------|
> > | Spatial-Feature Factorized Linear Readout (1)           | 0.83154 | 0.79296 | 0.7795 | 0.7419 | 0.7268 | 0.7323 | 0.7085 | 0.4847  | 0.4831 | 0.4504  |
> > | (1) + affine transforms only on spatial masks           | 0.8596  | 0.8217  | 0.8179 | 0.7769 | 0.7705 | 0.7719 | 0.7659 | 0.5638  | 0.5962 | 0.5371  |
> > | (1) + affine transforms only on feature maps            | 0.8750  | 0.8409  | 0.8310 | 0.7948 | 0.7814 | 0.7958 | 0.7782 | 0.5865  | 0.6156 | 0.5641  |
> >
> > *References continued in next comment.*

---

> ### Author Response · Authors · 2024-11-19
> **References**
>
> 1. Khosla, Meenakshi, and Leila Wehbe. "High-level visual areas act like domain-general filters with strong selectivity and functional specialization." bioRxiv (2022): 2022-03.
> 2. Joo, Jaehoon, Taejin Jeong, and Seongjae Hwang. "Brain-Streams: fMRI-to-Image Reconstruction with Multi-modal Guidance." arXiv preprint arXiv:2409.12099 (2024).
> 3. Liu, Yulong, et al. "See Through Their Minds: Learning Transferable Neural Representation from Cross-Subject fMRI." arXiv preprint arXiv:2403.06361 (2024).
> 4. Doerig, Adrien, et al. "Semantic scene descriptions as an objective of human vision." arXiv preprint arXiv:2209.11737 10 (2022).

---

> ### Comment · Reviewer_EMZd · 2024-11-23
>
> I appreciate the clarifications. I believe these additional details and analyses could strengthen the paper, but I do not believe have been incorporated (yet). I still have some concerns:
>
> You mention “the primary objective of our study was not to isolate these factors.” But as the paper is currently written, comparing neural versus task optimization seems central to the paper. In the abstract and throughout the paper “task optimized” and “response optimized” are directly contrasted to each other, and many conclusions explicitly mention this distinction.
>
> I appreciate the additional analysis with the Resnet-50 architecture. I think this would strengthen the paper and warrants some discussion. The lower performance of the “response optimized” model with this architecture however, further underscores my concern about the conclusions about “response” versus “task optimization” in the paper.
>
> The distinction between “local” and “global” semantics is interesting, but does not feel like a major advance over [4]. In fact, I believe that paper also compares word-level versus sentence-level embeddings).
>
> Finally, I appreciate the additional information about the STN motivation. I think this would significantly strengthen the paper, if added.

---

> ### Author Response · Authors · 2024-11-25
> **Updates for Reviewer EMZd**
>
> Thank you so much for your patience with us while we were updating our paper.  We wanted to mention the changes in the comments first as updating the paper took a little longer as we had to be careful with the 10 page limit constraint. Here are all the changes we have added throughout the paper based on the various suggestions (**Please refer to the latest revision of the paper**)-
>
> 1. Reorganizing Figure 1 based on the comments received by reviewers **gWb8** and **rgY2** -
>    - Added a brief overview of the entire pipeline (Schematic Pipeline of DNN Models)
>    - Separating out the individual components of the pipeline
>    - Replacing the tensor product symbol with reference to actual equation numbers used in the paper
> 2. Reorganized Figures - 2,3,4 by reducing the unused blank space in the cortical flatmaps according to reviewer **gWb8**.
> 3. Updated Figure 2B with discrete color maps as suggested by reviewers **rgY2** and **EMZd**.
> 4. Updated Figure 2A to highlight the advantages of STN readout as suggested by reviewer **EMZd**.
> 5. Better motivated the utility of Semantic Spatial Transformer Readout (**Lines 274-287**) based on the suggestions of reviewers **zmHZ**, **gWb8**, **rgY2** and **EMZd**.
>     - Added further analysis in **Supplementary section A.4 (Table 5, Figure 10)**, and they have been referenced in the main text at **line 287** - *Analyzing Spatial Modulation of Receptive Fields in Visual Cortex: Insights from STN Readouts*.
>     - As suggested by reviewer **rgY2**, we added further analysis on the dependency of the Semantic Spatial Transformer Readouts on channel size (and bias introduced by the readouts) in **Supplementary section A.5 (and Table 6)** - *Dependency of Semantic Spatial Transformer Readout on Channel Size*. This has been referenced in the main text in **line 333-334**.
> 6. As suggested by reviewer **zmHZ**, we added further analysis on the utility of dense captions in **Supplementary Section A.3 (and Figure 9)**, and referenced them in the main text at line **215**.
> 7. As suggested by Reviewer **EMZd**, we added further analysis on *‘Comparing Architectural Approaches for Task and Response Optimized models’* in **Supplementary Section A.6 and Table 7**, and referenced them in the main text at line **192**.
>
> **To be more specific, specific updates with respect to your comments are addressed at** -
>
> 1. **Questions** - Figure 2B is now updated with discrete colors at page 7
> 2. **Weakness 1**
>   - we added further analysis on Comparing Architectural Approaches for Task and Response Optimized models in Supplementary Section A.6 and Table 7, and referenced them in the main text at line **192**.
> 3. **Weakness 3**
>   - Figure 2A is updated to further highlight the performance boosts obtained by using the STN readout.
>   - Further clarification on the motivation behind the use of STN readouts is added at lines 274-287.
>   - Further analysis on STN readouts are added in Supplementary section A.4 (Table 5, Figure 10), and they are referenced in the main text at line 287
>
> **We understand that the reviewer still has some pending concerns regarding Weakness 1 and 2, and we shall respond to them very soon in a later comment.**

---

> > ### Comment · Reviewer_EMZd · 2024-11-25
> >
> > Thank you. I have updated my score.
> >
> > In addition to the points noted above, one other small suggestion is to frontload the biological relevance for STN in the motivation, if the authors feel that is indeed an important element.

---

> > > ### Author Response · Authors · 2024-11-28
> > > **Second Official Response to Reviewer EMZd**
> > >
> > > First of all we want to thank the reviewer for their continued suggestions to make our paper better.
> > >
> > > **Regarding Response and Task Optimized Models -**
> > >
> > > We once again want to highlight the distinction between ‘task’ and ‘response’ optimised models. We agree that __‘comparing neural versus task optimization’__ is central to the paper. We also agree that __“task optimized” and “response optimized” are directly contrasted to each other, and many conclusions explicitly mention this distinction.__
> > >
> > > To reiterate once more, task optimized models consist of a pretrained CNN core that has been pre trained extensively on large datasets such as ImageNet for very specific object oriented tasks. Only the final layer of these models are fine tuned to predict brain responses.  While it would be insightful to explore how different task optimized models pretrained on various tasks would perform (as already mentioned in discussion), such a comparison is beyond the scope of this study. Further, past research Conwell et al. (2022) have shown that varying architectures (convolutional vs non-convolutional such as transformers) pretrained on different training objectives have minimal impact on brain predictivity.
> > >
> > > However, as impressive as these models are, they are extremely biased towards the specific tasks that they have been trained on and hinder novel discoveries regarding the brain. To overcome this, researchers came up with Response Optimized models trained from scratch on brain data, with approximately 0.03 times the data that the task optimized models are trained on. The architecture needs to have a lot of built-in structure to model the brain as closely as possible, especially since the training diet is so small, as shown by Khosla and Wehbe.
> > >
> > > We argue that directly comparing response-optimized and task-optimized models, even with matched architectures, is challenging due to differences in training data—response-optimized models are trained on NSD (MS-COCO images), while task-optimized models use ImageNet. In addition to our previous experiments in Supplementary Table 7, we have now added two additional experiments with a Mask-RCNN encoder - (1) a task-optimized encoder pretrained for object detection on MSCOCO dataset, and (2) a response-optimized encoder trained from scratch on NSD dataset (which uses MSCOCO images, although a small subset of them). This further proves that merely training any architecture (e.g., AlexNet, ResNet-50, or transformers) on neural data does not make it inherently suited for response optimization. Thus, the difference between task- and response-optimized models goes beyond whether one is pre trained or not. Instead, it’s about selecting the most effective pretrained model (with optimized architecture and dataset) for task-optimized applications versus the most suitable architecture for response-optimized models that aligns with neural data constraints.
> > >
> > > **Regarding the differences between our paper with  Doerig et al. -**
> > >
> > > Building on our earlier clarifications as to how our work is unique from Doerig et al., we emphasize that the distinction between __'local' and 'global'__ in our work differs fundamentally from the comparison between __word-level versus sentence-level embeddings__ discussed by Doerig et al. Our local vs. global comparison introduces a contrast between globalized semantic information (e.g., a caption summarizing the entire image) and localized information (e.g., individual captions for distinct regions of the image, combining spatial and semantic knowledge). In contrast, Doerig et al.'s word-level vs. sentence-level comparison focuses on varying degrees of globalized semantics, depending on the comprehensiveness of the embeddings. These two conceptual frameworks are distinct and address different aspects of analysis.

---

> > > > ### Comment · Reviewer_EMZd · 2024-12-01
> > > >
> > > > Thank you for clarifying the 'local' / 'global' distinction refers to spatial scale. The distinction to Doerig et al., is more clear, but now I wonder if this finding is somewhat trivial given the increase in RF size from low- to high-level visual regions?

---

> > > > > ### Author Response · Authors · 2024-12-01
> > > > > **Third Official Response to Reviewer EMZd**
> > > > >
> > > > > Thank you for raising this important point. While receptive field size increases from low- to high-level visual regions, our findings highlight that the intermediate regions are not solely better modeled by purely visual features, as some prior studies using single captions have concluded (e.g., Doerig et al.). Instead, our analysis using dense captions reveals that these regions are also well-modeled by visual features that align with linguistic descriptions. This demonstrates a nontrivial interaction between visual and semantic alignment that is not captured by previous approaches relying on sparse or single-caption annotations.

---

### Official Review · Reviewer_gWb8 · 2024-11-04

**Soundness:** 3
**Presentation:** 1
**Contribution:** 3
**Rating:** 3
**Confidence:** 4

**Summary:**

Using fMRI data from the Natural Scenes Dataset, the authors investigate how different encoder backbones and readout mechanisms predict neural responses in the human visual system.

They compare a range of models, including those optimized for visual recognition (e.g., AlexNet, ResNet), neural response prediction, and language or vision-language tasks (e.g., CLIP, MPNET). Furthermore, they explore various readout mechanisms to map model activations to fMRI signals, introducing a novel approach (in the context of fMRI encoders) called the Semantic Spatial Transformer readout.

They find that:

1. Response-optimized models perform best in early visual areas (V1-V4): This suggests that these areas prioritize perceptual features not readily captured by linguistic descriptions or task-specific training.

2. Task-optimized and language models do better in higher visual areas: This indicates a shift towards semantic processing in these regions. Large language model embeddings, particularly those using detailed contextual descriptions, prove highly effective.

3. Semantic Spatial Transformer readout improves performance: This novel readout consistently outperforms existing methods like linear, Gaussian, and factorized readouts, boosting accuracy by 3-23%. This improvement stems from its ability to learn stimulus-specific modulations of receptive fields and feature maps.

**Strengths:**

I think overall, the authors' thorough experimentation is the greatest strength of this paper:

* **Systematic Comparison:** They do a reasonably systematic comparison, comparing a diverse range of models and readout mechanisms, which offers valuable insights.
* **Novel Readout Mechanism:** They propose (in the context of fMRI encoders) a novel readout mechanism—the using the previously proposed spatial transformer with differentiable bilinear sampling —and show that it indeed improves prediction accuracy.  This is a significant contribution in the context of fMRI encoders.
* **Identification of Cortical Regions:** They identify three cortical regions, largely aligned with prior hypotheses about visual cortical functionality. This further strengthens existing theories.
* **Good Discussion of Prior Work:** The authors do a reasonably good job in discussing prior fMRI encoder literature, effectively contextualizing their research.  This demonstrates a solid understanding of the field.

**Weaknesses:**

The presentation of this paper could be *significantly* improved. I think the presentation quality of this paper does not match the quality of other ICLR papers I am currently reviewing or have reviewed in past years, or ICLR papers that have been accepted in prior years.

The figures are unclear and lack consistent formatting, notation often unexplained, and significant wasted space.

My specific concerns are below:
1. Figure 1 -> This figure is very cluttered and very confusing. Why are the subfigure legends (A, and B) placed so randomly?
2. Figure 1 -> What is the 'What' weight matrix, where does it come from? It is obvious if you are familiar with fMRI encoder literature, but describing the interaction between the weight matrix and green using a tensor product $\otimes$ seems very misleading. This tensor product is also used in the upper part as well, which is deeply misleading, and instead should be expressed as a transposed matrix product. These symbols have a meaning and without redefinition this is pretty confusing.
3. Figure 1 -> Why is the task optimized framework placed together with the response optimized framework without any clarification of which is which?
4. Figure 1 -> Why is the dense captioning output also part of the response optimized framework with a rotation equivariant network?
5. Where are the dense captions coming from? `Line 210` says `An image of size 424 ∗ 424 is divided into grids of size 53 ∗ 53`, but does not otherwise clarify the origin of the captions anywhere in the paper. What model is used here?
6. `Line 224` please avoid vector matrix products. This assumes row vectors which is not standard.
7. `Line 237` what are the shapes of the output of function $V_c(x,y)$? Could you describe this sampling in more detail?
8. Spatial transformer section, `Line 269` it is unclear what $\theta_2$ is. This is a really important part of the paper and a key claimed contribution. Could the authors mathematically clarify how $\theta_2$ plays a role?
9. `Line 286`, what is `AT`?
10. `Line 297` are the models not voxel wise models?
11. Figure 2A, the bottom descriptions of the models is very confusing. How is the "Language Encoder" being used with "Semantic Spatial Transformer Readouts"?
12. Figure 2B, please use a proper categorical color map for discrete data.
13. Minor -- Figure 3B, extra space before optimized
14. Figure 3, how are you defining "better predicted by Task Optimized"? Is this the best of ResNet50 or AlexNet? Why use these models when CLIP has been shown to be the best model of visual responses?
15. All flatmaps have significant wasted space.
16. Lack of analysis of the spatial transformer networks. While the paper claims STNs as a significant contribution, there is no visualization of the affine parameters for each voxel. Do the affine parameters focus on population receptive fields that are provided in NSD?

**Questions:**

Please see weaknesses section for the questions.

---

> ### Author Response · Authors · 2024-11-23
> **Official Response to Reviewer gWb8**
>
> Thank you for taking the time out to review our paper. We apologize for some of the inconsistencies in our presentation, and here are some of our fixes, and further clarifications as per your comments in the Weakness section -
>
> **Addressing concerns regarding Figure 1** - Figure 1 is the introductory figure which gives a brief overview of our experimental pipelines. We have dealt with three kinds of model inputs in our experiments - raw image data, single caption giving a brief overview of the entire image and dense captions giving detailed information about various subparts of the image. Raw pixel inputs (of shape 3*256*256) are passed through CNN encoders and caption stimuli are passed through LLM encoders and then a readout layer is used to map the encoder feature maps to brain voxel responses. The CNN encoder can be either a pre-trained task optimized model or a response optimized model that is trained from scratch. The readout layers can be a linear ridge regression readout, a gaussian 2D readout, a Spatial Feature Factorized readout or a Semantic Spatial Transformer readout. The single caption stimuli are directly passed to only the linear ridge regression readout as they contain no spatial information.
>
> - **Weakness 1 -** In the revised version of our paper, we added boxes separating each of the readouts and input stimuli, and also aligned the subfigure legends. We would be happy to make the figure more intuitive as per reviews.
>
> - **Weakness 2 -** We apologize for not making the ‘what’ and ‘where’ matrix more clear in our paper. We have added more details regarding this under Spatial-Feature Factorized Linear Readouts in section 2.2, specifically at lines 242, 243, 251 and 252. We also apologize for using the tensor product representation in our figure. IN the current revised version, we have replaced them with more suitable changes according to the actual operation they perform.
>
> - **Weakness 3 -** As mentioned earlier, the encoder can either be task or response optimized. Our framework is flexible in that way.
>
> - **Weakness 4 -** As mentioned earlier, the dense caption outputs are also passed through a CNN core (though it is just a single convolutional block), and it is explained in more details under Language Models -> Dense Captions in section 2.1, specifically in lines 210-215.
>
> **We will soon post another revision with a more updated version of Figure 1 for further clarity.**
>
> **Addressing Weakness 5 -** The dense captions are generated using GPT-2. This is already mentioned under Language Models -> Dense Captions in section 2.1, specifically in line 211.
>
> **Addressing Weakness 6 -** This has now been updated in our current revision. Yi is now a column vector, and Ei is also a column vector.
>
> **Addressing Weakness 7 -** We apologize for not being more clear with the function $V_c(x,y)$.
> Let’s say we have a feature map of shape $C\*H\*W$ from an encoder where $C$ is the number of channels, and $H$ and $W$ are the dimensions of each channel.  Each voxel learns a Gaussian distribution across the receptive field, the mean of which is the location the voxel is most sensitive to. For each voxel, lets represent their receptive field center by $(x,y)$ which is the mean of the gaussian distribution for the respective voxel (this lies within $W\*H$). A value is chosen for each channel by sampling this gaussian distribution which is what is returned by 	$V_c(x,y)$. Its output shape is $C\*N\*1$, where $N$ is the number of voxels. The voxel response is further calculated by a weighted sum across $C$ (dimension 0) as explained in line 236.
> We were purposely very brief with it as it's part of an already published work [1], and if needed we can add more details in the appendix.
>
> **Addressing Weakness 8 -** $𝜃_2$ defines the six affine parameters per voxel, which are used to modulate spatial masks independently for each voxel. This design is inspired by neuroscience studies showing that spatial receptive fields (RFs) are not static but can dynamically adapt based on the stimulus. For instance, seminal research has demonstrated that RF sizes can shrink or expand depending on stimulus contrast (Sceniak et al., Nature Neuroscience, 1999) and can be modulated by contextual influences such as surround modulation or attentional shifts (Womelsdorf et al., Nature Neuroscience, 2006). Unlike other linear readouts, such as the Spatial-Feature Factorized Linear Readout or Gaussian 2D readout, which use fixed spatial masks irrespective of the stimulus, the Semantic Spatial Transformer enables stimulus-dependent modulation of RFs. By leveraging affine transformations, it can capture a range of spatial changes, including translation, scaling, rotation, and shearing.
>
> **References -**
>
> 1. Lurz, Konstantin-Klemens, et al. "Generalization in data-driven models of primary visual cortex." BioRxiv (2020): 2020-10.
>
> *Response continued in the next comment.*

---

> ### Author Response · Authors · 2024-11-23
> **Continued Official Response to Reviewer gWb8**
>
> **Addressing Weakness 9 -** $AT(X, \theta)$ is a function that performs an affine transformation (such as translation, rotation etc) on each channel of matric $X (C\*H\*W)$ ($C$=number of channels, $H\*W$=dimension of each channel) using a set of $C$ transforms $\theta$ ($C\*2\*3$), ie affine transform $\theta_i$ is applied to $X_i$. This has already been mentioned in lines 287-288.
>
> **Addressing Weakness 10 -** Yes the models are voxelwise models. We model the voxel responses for each region separately. We have clarified this in the revised statement in line 296  “We trained separate models for **voxels** in each of the following brain regions: the high-level ventral, dorsal and lateral streams, V4, V3v, V3d, V2v, V2d, V1v, and V1d.”
>
> **Addressing Weakness 11 -** Language encoder is a  shallow CNN (a single convolutional block) that processes dense language caption embeddings. Note that each dense caption is associated with a specific spatial region of the image, and the language encoder preserves this spatial layout, enabling integration with the Semantic Spatial Transformer Readout. This is already clarified in the image caption. Further details can be found in  Language Models -> Dense Captions in section 2.1.
>
> **Addressing Weakness 12 -** We apologize for this, and have updated this in our current revision.
>
> **Addressing Weakness 13 -** Thank you for pointing this out. This is fixed in the latest revision.
>
> **Addressing Weakness 14 -**We compare the performance of each voxel using a response optimized encoder and a task optimized encoder, both encoders using a semantic transformer readout. When we say ‘better predicted by task optimized’, we simply refer to whether the test performance of that voxel is better modeled by a task optimized model or not. We do not take the best of ResNet or AlexNet in this situation. We compare them individually with the response optimized encoders. In figure 3B, the top most plot compares the response optimized encoders with a resnet50 encoder, and the lower plot compares a response optimized encoder with an alexnet encoder. Due to timing constraints, we were only able to experiment with two of the models. While other computational objectives or architectures might improve predictivity, recent findings (e.g., Conwell et al. 2023) using the same NSD dataset demonstrate that qualitatively different architectures (e.g., CNNs vs. Transformers) and vastly different task objectives (e.g., purely visual contrastive learning vs. vision-language alignment as in CLIP) achieve comparable levels of brain predictivity. Thus, we do not anticipate that including the CLIP model would significantly alter the conclusions of our study.
>
> **Addressing Weakness 15 -** We apologize for this, and have updated this in our current revision.
>
>
> *Response continued in next comment.*

---

> > ### Author Response · Authors · 2024-11-24
> > **Continued Official Response to Reviewer gWb8**
> >
> > **Addressing Weakness 16 -** The affine parameters are not learnt on population receptive fields provided in NSD, and are independent of them. One of the major advantages of using STNs is that it only uses visual stimuli to outperform existing readouts that depend upon additional information such as retinotopic mapping or population receptive fields. The parameters of STN are learnt with a help of a pretrained image encoder that highlights the most important parts of an image for each voxel. To be more specific, the Semantic Spatial Transformer Readout applies affine transformations for each channel of the encoder feature representations and the spatial features (‘where’ matrix), which are learnt based on the ‘important’ parts of the image. We provide more motivation behind the STN readout below:-
> >
> > **a. Spatial Modulation of Spatial Masks (Spatial Receptive Fields):**
> > The STN allows voxel-specific spatial modulation of receptive fields (RFs), inspired by studies demonstrating the dynamic adaptability of RFs to stimulus properties:
> > - RF sizes can expand or contract based on contrast (Sceniak et al., Nature Neuroscience, 1999).
> > - RFs can also shift or reshape in response to contextual or attentional influences (Womelsdorf et al., Nature Neuroscience, 2006).
> >
> > Unlike fixed spatial masks used in other linear readouts (e.g., Spatial-Feature Factorized Linear or Gaussian 2D readouts), the STN employs affine transformations to capture stimulus-dependent changes in spatial receptive fields, such as translation, scaling, rotation, and shearing, in line with existing neuroscientific results. This flexibility enables more accurate modeling of neural responses that exhibit dynamic spatial RF properties.
> >
> > **b. Spatial Modulation of Feature Maps:** The STN also spatially transforms feature maps (i.e. the encoder channels). Each channel typically encodes distinct attributes (e.g., edges, textures, shapes), and the ability to apply channel-specific transformations allows the STN to adapt to their unique geometric properties. For instance, one channel may require scaling to emphasize fine-grained details, while another might need rotation for orientation invariance. This flexibility is particularly advantageous for neural response modeling. Unlike object classification tasks, where we can employ augmentations tailored to known invariances (e.g. rotating an image won’t change the category label) to boost predictive accuracy, the geometric invariances of voxel responses are unknown. STN enables the network to learn these invariances directly from the data, providing a crucial advantage in predicting voxel responses across diverse visual regions.
> >
> > **The above reasoning will be added in the next revision of our paper.**
> >
> > In an additional experiment focused on interpreting the STN readouts, we calculated the distance between the affine parameters corresponding to the spatial maps of each voxel for every image, relative to the mean affine parameters across all images. The L2 norm of this vector was computed for each voxel. Across all encoders, we observed that stimulus-dependent spatial shifts of the receptive field increase from lower to higher visual regions. A similar trend emerged when calculating the average spatial shifts for each channel of the feature map across images for different regions. This trend further supports the idea that higher levels of the visual cortex benefit more from learned geometric invariances and exhibit greater spatial modulation of their visual receptive fields compared to lower visual cortex regions. This modulation includes phenomena such as receptive field expansion, contraction, or shifts in response to different stimuli.
> >
> > **Please see Supplementary Fig. 10 in our revised paper for the results of this analysis. This analysis has been added in Supplementary section A.4 in the revised version of our paper.**

---

> > > ### Comment · Reviewer_gWb8 · 2024-11-25
> > >
> > > I wrote a response and then deleted it. Because it does not seem like the author's comments reflect the current state of the paper. I would be happy to take another look once the author's revision reflect their comments to me.

---

> ### Author Response · Authors · 2024-11-25
> **Updates for Reviewer gWb8**
>
> Thank you so much for your patience with us while we were updating our paper.  We wanted to mention the changes in the comments first as updating the paper took a little longer as we had to be careful with the 10 page limit constraint. We apologize for the version inconsistencies for which you could not see our latest updates. Here are all the changes we have added throughout the paper based on the various suggestions (**Please refer to the latest revision of the paper**)-
>
> 1. Reorganizing Figure 1 based on the comments received by reviewers **gWb8** and **rgY2** -
>    - Added a brief overview of the entire pipeline (Schematic Pipeline of DNN Models)
>    - Separating out the individual components of the pipeline
>    - Replacing the tensor product symbol with reference to actual equation numbers used in the paper
> 2. Reorganized Figures - 2,3,4 by reducing the unused blank space in the cortical flatmaps according to reviewer **gWb8**.
> 3. Updated Figure 2B with discrete color maps as suggested by reviewers **rgY2** and **EMZd**.
> 4. Updated Figure 2A to highlight the advantages of STN readout as suggested by reviewer **EMZd**.
> 5. Better motivated the utility of Semantic Spatial Transformer Readout (**Lines 274-287**) based on the suggestions of reviewers **zmHZ**, **gWb8**, **rgY2** and **EMZd**.
>     - Added further analysis in **Supplementary section A.4 (Table 5, Figure 10)**, and they have been referenced in the main text at **line 287** - *Analyzing Spatial Modulation of Receptive Fields in Visual Cortex: Insights from STN Readouts*.
>     - As suggested by reviewer **rgY2**, we added further analysis on the dependency of the Semantic Spatial Transformer Readouts on channel size (and bias introduced by the readouts) in **Supplementary section A.5 (and Table 6)** - *Dependency of Semantic Spatial Transformer Readout on Channel Size*. This has been referenced in the main text in **line 333-334**.
> 6. As suggested by reviewer **zmHZ**, we added further analysis on the utility of dense captions in **Supplementary Section A.3 (and Figure 9)**, and referenced them in the main text at line **215**.
> 7. As suggested by Reviewer **EMZd**, we added further analysis on *‘Comparing Architectural Approaches for Task and Response Optimized models’* in **Supplementary Section A.6 and Table 7**, and referenced them in the main text at line **192**.
>
> **To be more specific, specific updates with respect to your comments are addressed at** -
>
> 1. **Weakness 1** - Figure 1 has now been updated and can be found at page 3 of the current revision.
> 2. **Weakness 6** - This has now been updated in our current revision. $Y_i$ is now a column vector, and $E_i$ is also a column vector. These updates can be found at line 223-224
> 3. **Weakness 8** - Please refer to the further clarifications on this in the previous comments. More reasoning on this is added in the main text from lines 274-287. We also added further analysis on the Semantic Spatial Transformer Readouts and the implications of $\theta_1$ and $\theta_2$ in Supplementary Section A.4, Figure 10 and Table 5.
> 4. **Weakness 12, 13, 15** - These figures (2,3,4,5) have been updated in our latest revision.
> 5. **Weakness 16** - Please refer to the further clarifications on this in the previous comments. More reasoning on this is added in the main text from lines 274-287. We also added further analysis on the Semantic Spatial Transformer Readouts and the implications of $\theta_1$ and $\theta_2$ in Supplementary Section A.4, Figure 10 and Table 5.
>
> **No changes were needed for Weakness 2,3,4,5,7,10,11 and 14. Please refer to above comments for clarifications regarding the same.**
>
> Due to edits in the paper, following changes can be seen for the below weaknesses -
>
> 1. **Weakness 9** - Please refer to the above comments for further clarification. Line 287-288 (in the above comment) has now been shifted to line 272-273
>
> **Please let is know if you still see inconsistencies in the current version. We can raise appropriate concerns if the current version as seen by the authors is not being reflected on the reviewer's end.**
>
> *We would be happy to address any further concerns that you have.*

---

> ### Author Response · Authors · 2024-12-02
> **Request for Review of Updated Paper and Revisions (Reviewer gWb8)**
>
> Dear reviewer gWb8,
>
> Thank you for your thoughtful and detailed comments, especially regarding clarity and presentation, which have greatly enhanced our work. We have addressed all your suggestions in the revised manuscript (see the previous comment for details). If possible, could you review the updated version and let us know if you have any additional concerns? Today marks the end of the discussion phase. Thank you again for your valuable input.
>
> Thanks,
>
> Authors.

---

> ### Comment · Reviewer_gWb8 · 2024-12-02
>
> I thank the authors for the rebuttal and the revision.
>
> I took a look at the revised PDF, in my view the current revision is still pretty flawed.
>
> 1. `Objecttives`?
> 2. Top left of Figure 1 -- are the neurons the input or output?
> 3. Line 223 -> The issues I mentioned in the original review still remain
>
> I maintain my score of 3 and I strongly recommend against acceptance.

---

> > ### Author Response · Authors · 2024-12-03
> > **Further Response to Reviewer gWb8**
> >
> > We sincerely thank the reviewer for their continued feedback and for taking the time to review our rebuttal and revised manuscript.
> >
> > We acknowledge and apologize for the following typographical errors in the current revision:
> >
> > 1. In Figure 1A, the word "Objectives" is misspelled.
> > 2. The schematic diagram in the top left of Figure 1A incorrectly uses a downward arrow. The arrow should point upward, indicating the flow from the “Readout Layer” to “N Neurons.”
> >
> > We deeply appreciate the reviewer bringing these issues to our attention. These are typographical errors that can be easily corrected in the camera-ready version. While we respect the reviewer’s concerns, we hope it is clear that these minor issues do not compromise the scientific integrity of the work, and we feel that labeling the paper as "pretty flawed" due to these errors might not fully reflect its overall quality and contributions.
> >
> > Regarding point 3, the reviewer previously noted that “please avoid vector matrix products. This assumes row vectors which is not standard.” In response, we revised the notation in line 223 to use $E_i$ and $Y_i$ as column vectors. This approach is mathematically correct and aligns with established conventions. As such, we believe this is largely a stylistic preference rather than a substantive error.
> >
> >
> > While we respect the reviewer's assessment, we feel that the current score may not fully reflect the strengths of the paper, including the contributions that the reviewer acknowledged in their original feedback. We appreciate the opportunity to clarify these points and remain hopeful that the revised manuscript demonstrates the merit of our work.

---

### Official Review · Reviewer_zmHZ · 2024-11-04

**Soundness:** 3
**Presentation:** 2
**Contribution:** 4
**Rating:** 6
**Confidence:** 4

**Summary:**

In this work, the authors present a comprehensive suite of analyses comparing vision / language DNN models to human fMRI data. Using a novel “readout” mechanism designed explicitly to account for space in the mapping of DNN embeddings to brain activity, the authors report localizing 3 sub-regions in the human visual cortex that respond differentially to spatial and semantic information.

**Strengths:**

The use of deep neural network models to predict and understand the structure of representation in the biological visual system is a practice rife with heretofore unanswered, but deeply foundational questions as to how it should be done. Bucking a trend that far too often recycles canonical, but relatively unscrutinized methods to new models or new brain data, this submission is impressive not just for the fact that it tackles these questions head-on, but tackles so many of them simultaneously -- and does so (mostly) without losing the forest for the trees. For this alone, I applaud the authors and can recommend that this paper be accepted.

**Weaknesses:**

My major concern here (and one that I admit is not fully within the authors control, but which clarifying updates or different narrative focus could nonetheless address) is the lingering doubt as to whether even these newer, more expertly designed methods actually do give us any meaningful new “insights” about the biological system they’re nominally designed to give us insights about. An overly reductionist summary of the “findings” of this analysis with respect to the human visual brain could well be that they simply provide more evidence for what is already a amply established gradient of increasingly “abstract” visual information from early (more view-dependent) areas (where smaller, localized receptive fields and retinotopy are the dominant representational motifs) to later (less view-dependent) visual areas (where -- depending on which side of the ventral / dorsal divide those areas are closer to -- you begin to get “representations” that evoke “object categories”, “navigational affordances”, or “conceptual semantics”). And while much debate does remain as to many of the details here, it seems (to me at least) that the existence of this gradient is more or less a common consensus.

**Questions:**

My primary suggestion for strengthening this paper is more or less solely for the authors to lean further into its greatest strength -- and to further explicate or justify the expert methods that differentiate this work from so many others attempting to tackle similar questions. Needless to say, perhaps, there are a number of ways the authors could do this. Below are a few different “options” that (I hope) seem reasonable given the constraints of the current review. The authors should feel free to choose however many / whichever of these seems most feasible or intuitive. For me, at least, almost any movement along these vectors is movement that would increase the value of this work for the target audience it seems intended for:
- “Beyond accuracy”: The primary justification of the authors’ “novel readout mechanism” is the general increase in accuracy it provides over other methods. But the emphasis on accuracy as the primary advantage rings a bit hollow if a major part of the goal here is to gain insight into the structure of  representation in biological cortex. There are many alternative ways (e.g. data augmentation, denoising, nonlinearities) -- even “hacky” ones (e.g. smaller cross-validation splits) that one could use to increase the predictive accuracy of model readout mechanisms. What demonstrable advantage does the “semantic spatial transformer” readout have over readout methods with respect to the theoretical questions at play here? (An example answer: “ordinal least squares or regularized regression-style readouts do not preserve spatial information -- therefore making certain areas of the brain appear to be more transform-invariant than they likely are in reality. Here’s a metric that operationalizes the probability of transform-invariance in the fMRI data without models. And here is a side-by-side comparison of the transform-invariance we estimate with ridge regression and our STN readout, respectively.”)
- “Semantics” without language model confounds: There are a number of issues (again, beyond the scope of this paper, but nonetheless relevant) with the use of language models as predictive models of visual fMRI data -- including the fact the inputs to these models (tokenized words) are already proto-symbolic at the time of their initial injection into the candidate representational models that embed them (and are thus more abstract by default than the pixels injected into vision models); and also, an increasing “convergence” between vision and language models [1] that suggests a sort of “default” alignment between these systems attributable (most likely, it seems) to biases in their training data. How to get at questions of “semantics” without over-interpreting language models? One way, perhaps, is to reconsider the brain data itself: It has been suggested by [2] that certain kinds of transformations on the underlying brain data to be modeled (e.g. aggregating across multiple neurons in the case of electrophysiology) can instantiate properties like linearly-separable category boundaries not otherwise apparent without those transformations. If something like this is done on the features of vision models (e.g. averaging across multiple images of the same visual concept), do vision models begin to look more “semantic”? Perhaps the semantic spatial transformer could be used to unveil precisely the kinds of feature transformations that occur along the gradient from early to late visual cortex.
- “densifying” the “single” captions: The authors claim that the localized semantic descriptions inherent to their “dense” captioning method unveil a noticeable midpoint between early, more spatiotopic representations and later, “globally” abstract representations. But is this really about the local tagging of an image’s subparts? Providing more comprehensive “single captions” of the full image that includes more extensive specification of details might close the gap between the dense captioning method and the global captioning -- but in a way that obviates the need to manually subdivide the image. In short, adding further detail (with and without explicitly spatial language) seems like an important control for downstream interpretation of this result.

[1] Huh, M., Cheung, B., Wang, T., & Isola, P. (2024). The platonic representation hypothesis. arXiv preprint arXiv:2405.07987.
[2] Vinken, K., Prince, J. S., Konkle, T., & Livingstone, M. S. (2023). The neural code for “face cells” is not face-specific. Science Advances, 9(35), eadg1736.

---

> ### Author Response · Authors · 2024-11-22
> **Official Response to Reviewer zmHZ (Question: Beyond Accuracy)**
>
> Thank you so much for taking the time out to review our paper and for suggesting ways to improve our experiments. We would add all of the below analysis to the supplementary section of our paper in a later revision.
>
> **“Beyond Accuracy” -** This is an important point, and we appreciate the reviewer highlighting it. We do feel that achieving high accuracy can be a very useful goal in many contexts when developing models of the human brain. For e.g., accurate and robust neural network models are essential for in-silico experiments [2] to explore hypotheses that would be challenging or impractical to test in vivo, inform experimental design, as well as for precise neural population control. In this way, accuracy provides a foundational utility for practical and theoretical advancements.
>
> **Further diving into the question - What demonstrable advantage does the “semantic spatial transformer” readout have over readout methods with respect to the theoretical questions at play here?**
>
> We also want to emphasize that the STN readout is not merely a hack  to boost prediction accuracy;  its design is well-grounded in biological principles. We expand on this below. We first emphasize that the STN does not involve unnecessarily complex transformations. Instead, it only *spatially modulates* the feature maps and spatial masks, allowing for dynamic and stimulus-dependent adjustments.
>
> - **Spatial Modulation of Spatial Masks (Spatial Receptive Fields):**
> The STN allows voxel-specific spatial modulation of receptive fields (RFs), inspired by studies demonstrating the dynamic adaptability of RFs to stimulus properties:
>    - RF sizes can expand or contract based on contrast (Sceniak et al., Nature Neuroscience, 1999).
>    - RFs can also shift or reshape in response to contextual or attentional influences (Womelsdorf et al., Nature Neuroscience, 2006).
>
>    Unlike fixed spatial masks used in other linear readouts (e.g., Spatial-Feature Factorized Linear or Gaussian 2D readouts), the STN employs affine transformations to capture stimulus-dependent changes in spatial receptive fields, such as translation, scaling, rotation, and shearing, in line with existing neuroscientific results. This flexibility enables more accurate modeling of neural responses that exhibit dynamic spatial RF properties.
> - **Spatial Modulation of Feature Maps:**
> The STN also spatially transforms feature maps (i.e. the encoder channels). Each channel typically encodes distinct attributes (e.g., edges, textures, shapes), and the ability to apply channel-specific transformations allows the STN to adapt to their unique geometric properties. For instance, one channel may require scaling to emphasize fine-grained details, while another might need rotation for orientation invariance. This flexibility is particularly advantageous for neural response modeling. Unlike object classification tasks, where we can employ augmentations tailored to known invariances (e.g. rotating an image won’t change the category label) to boost predictive accuracy, the geometric invariances of voxel responses are unknown. STN enables the network to learn these invariances directly from the data, providing a crucial advantage in predicting voxel responses across diverse visual regions. Below is an additional experiment showing the relative improvements of the STN with only affine parameters for the spatial mask vs affine parameters for only the feature maps.  We see that both contribute to the improved performance of the STN readout.
>
> | Readout Type (all with response optimized E2cnn encoder) | V1v    | V1d    | V2v    | V2d    | V3v    | V3d    | V4     | Ventral | Dorsal | Lateral |
> |----------------------------------------------------------|--------|--------|--------|--------|--------|--------|--------|---------|--------|---------|
> | Spatial-Feature Factorized Linear Readout (1)           | 0.83154 | 0.79296 | 0.7795 | 0.7419 | 0.7268 | 0.7323 | 0.7085 | 0.4847  | 0.4831 | 0.4504  |
> | (1) + affine transforms only on spatial masks           | 0.8596  | 0.8217  | 0.8179 | 0.7769 | 0.7705 | 0.7719 | 0.7659 | 0.5638  | 0.5962 | 0.5371  |
> | (1) + affine transforms only on feature maps            | 0.8750  | 0.8409  | 0.8310 | 0.7948 | 0.7814 | 0.7958 | 0.7782 | 0.5865  | 0.6156 | 0.5641  |
>
>
> *Response to "Beyond Accuracy" continued in the next comment.*

---

> ### Author Response · Authors · 2024-11-22
> **Continued Official Response to Reviewer zmHZ (Question: Beyond Accuracy)**
>
> In an additional experiment focused on interpreting the STN readouts, we calculated the distance between the affine parameters corresponding to the spatial maps of each voxel for every image, relative to the mean affine parameters across all images. The L2 norm of this vector was computed for each voxel. Across all encoders, we observed that stimulus-dependent spatial shifts of the receptive field increase from lower to higher visual regions. A similar trend emerged when calculating the average spatial shifts for each channel of the feature map across images for different regions. This trend further supports the idea that higher levels of the visual cortex benefit more from learned geometric invariances and exhibit greater spatial modulation of their visual receptive fields compared to lower visual cortex regions. This modulation includes phenomena such as receptive field expansion, contraction, or shifts in response to different stimuli.
>
> **Please see Supplementary Fig. 10 in our revised paper for the results of this analysis. This analysis has been added in Supplementary section A.4 in the revised version of our paper.**
>
> *Responses to other questions continued in the next comment*

---

> ### Author Response · Authors · 2024-11-22
> **Continued Official Response to Reviewer zmHZ (Question: Semantics” without language model confounds and “Densifying” the single captions)**
>
> **“Semantics” without language model confounds -**  We appreciate the reviewer’s insightful comments. We acknowledge that language models inherently operate on proto-symbolic inputs (e.g., tokenized words) that are more abstract than the raw pixel inputs provided to vision models. This abstraction likely facilitates alignment with higher-level semantic representations in the brain, even if it does not directly inform the structure of visual cortex representations. Additionally, the growing overlap between vision and language models, especially as models get larger, may result in "default" alignments of the language models with visual cortex responses that do not provide any meaningful insights into the structure of visual cortex representations.
>
> To address the risk of overinterpretation, we have revised the discussion in the manuscript to caution against equating language models’ predictive success with evidence that the visual cortex encodes language-like representations (Line 532-536). Instead, we suggest that such success may reflect the ability of language models to capture the statistical structure of the visual world, given their training on vast datasets. This interpretation is consistent with recent findings that language model representations can also predict visual cortex activity in macaques—a species without linguistic abilities—underscoring that their predictive power may not depend on linguistic abstraction per se.
>
> The question of whether and how vision models develop “semantic” representations is indeed compelling but falls beyond the scope of the present study. Aggregating representations across multiple images of the same concept (e.g., different pictures of dogs) could potentially enhance semantic distinctions in vision models. This line of inquiry would be a promising direction for future work investigating the emergence of semantic representations in vision models and their relationship to language models.
>
> Regarding the reviewer’s final point, the Semantic Spatial Transformer (SST) operates by spatially transforming feature maps and spatial masks to model dynamic, stimulus-dependent adjustments, without fundamentally altering the nature of the features themselves. While this limits its ability to reveal specific feature transformations along the cortical hierarchy, future work could leverage STNs to explore the geometric invariances underlying neural responses in different brain regions. This remains an exciting avenue for further exploration.
>
> **“Densifying” the single captions -** This is an excellent point made by the reviewer about whether the observed differences between dense and global captioning are due to (a) the spatial subdivision of the image (Hypothesis 1) or the increased semantic detail in dense captions (Hypothesis 2). The original idea behind using dense captions was to provide spatial information in addition to semantic information in the form of captions, and subdividing the image into equal sized grids and getting captions for each grid was one of the easiest and most intuitive ways to do that. We also played around with other options such as using the DenseCap [1] model to identify parts of the image and generate captions for them. However, there were some regions which remain unidentified, and in comparison the dense captioning method proposed in the paper performed better.
>
> As per some of your suggestions, we tried generating more comprehensive single captions of the image using existing LLMs, however none of them were able to provide more information than those already present in the original MS-COCO dataset. In an attempt to densify the single captions, we thus adopted a different approach: for each image, we took the embeddings of dense captions generated for individual grid locations and averaged these embeddings to produce a single "aggregate dense caption" embedding.
>
> On comparing single caption stimuli with ‘densified’ single caption stimuli (as opposed to the dense caption approach discussed in the paper), we saw a similar trend where the higher regions of the visual cortex were better modeled by single caption stimuli. However, the transition in sensitivity from dense to single caption in the middle regions of the ventral, dorsal and lateral stream that is so clearly pronounced when using dense captions is missing when using the above ‘densified’ single captions. Further comparing ‘densified’ single captions to dense captions (as proposed in the paper), we saw that the dense captions modeled the overall visual cortex better. Hence, we do feel that adding spatial information to the dense caption is necessary for building more accurate models, be it by sub-dividing the image into grids or via any other way. **Please see Supplementary Fig. 9 in our revised paper for the results of this analysis. This analysis has been added in Supplementary section A.3 in the revised version of our paper.**

---

> ### Author Response · Authors · 2024-11-25
> **Updates for Reviewer zmHZ**
>
> Thank you so much for your patience with us while we were updating our paper.  We wanted to mention the changes in the comments first as updating the paper took a little longer as we had to be careful with the 10 page limit constraint. Here are all the changes we have added throughout the paper based on the various suggestions (**Please refer to the latest revision of the paper**)-
>
> 1. Reorganizing Figure 1 based on the comments received by reviewers **gWb8** and **rgY2** -
>    - Added a brief overview of the entire pipeline (Schematic Pipeline of DNN Models)
>    - Separating out the individual components of the pipeline
>    - Replacing the tensor product symbol with reference to actual equation numbers used in the paper
> 2. Reorganized Figures - 2,3,4 by reducing the unused blank space in the cortical flatmaps according to reviewer **gWb8**.
> 3. Updated Figure 2B with discrete color maps as suggested by reviewers **rgY2** and **EMZd**.
> 4. Updated Figure 2A to highlight the advantages of STN readout as suggested by reviewer **EMZd**.
> 5. Better motivated the utility of Semantic Spatial Transformer Readout (**Lines 274-287**) based on the suggestions of reviewers **zmHZ**, **gWb8**, **rgY2** and **EMZd**.
>     - Added further analysis in **Supplementary section A.4 (Table 5, Figure 10)**, and they have been referenced in the main text at **line 287** - *Analyzing Spatial Modulation of Receptive Fields in Visual Cortex: Insights from STN Readouts*.
>     - As suggested by reviewer **rgY2**, we added further analysis on the dependency of the Semantic Spatial Transformer Readouts on channel size (and bias introduced by the readouts) in **Supplementary section A.5 (and Table 6)** - *Dependency of Semantic Spatial Transformer Readout on Channel Size*. This has been referenced in the main text in **line 333-334**.
> 6. As suggested by reviewer **zmHZ**, we added further analysis on the utility of dense captions in **Supplementary Section A.3 (and Figure 9)**, and referenced them in the main text at line **215**.
> 7. As suggested by Reviewer **EMZd**, we added further analysis on *‘Comparing Architectural Approaches for Task and Response Optimized models’* in **Supplementary Section A.6 and Table 7**, and referenced them in the main text at line **192**.
>
> **To be more specific, specific updates with respect to your comments are addressed at** -
>
> 1. **“Beyond accuracy”** -
>   - Further clarification on the motivation behind the use of STN readouts is added at lines 274-287.
>   - Further analysis on STN readouts are added in Supplementary section A.4 (Table 5, Figure 10), and they are referenced in the main text at line 287.
> 2. **“Densifying” the single captions** - we added further analysis on the utility of dense captions in Supplementary Section A.3 (and Figure 9), and referenced them in the main text at line 215.

---

> > ### Author Response · Authors · 2024-12-02
> > **Request for Review of Updated Paper and Revisions (Reviewer zmHZ)**
> >
> > Dear reviewer zmHZ,
> >
> > Thank you for your thoughtful and detailed comments, particularly your insightful suggestions, which have significantly improved our work. We have carefully addressed all your concerns, implemented two of your major suggestions in the revised manuscript, and incorporated a discussion of your third suggestion in the discussion section (please refer to the previous comment for details). If possible, could you review the updated version and let us know if you have any additional concerns? Today marks the end of the discussion phase. Thank you again for your valuable input.
> >
> > Thanks,
> >
> > Authors.

---

### Author Response · Authors · 2024-11-30
**Official Comment to Area and Program Chairs**

We want to thank everyone for taking the time out to review our paper. First, we want to summarize the major contributions of our paper in the domain of modeling the human visual stream using DNNs as follows -

1. **Introduction of a novel readout mechanism:** We introduce a novel readout method utilizing Spatial Transformers, which delivers significant improvements in accuracy (3-23%) compared to previously employed SOTA readouts.

2. **Identification of brain regions responsive to visual and semantic input:** Through comparative analysis of models across various visual regions, we identify three distinct regions in the human visual cortex that respond primarily to (a) perceptual characteristics of the input, (b) localized visual semantics aligned with linguistic descriptions, and (c) global semantic interpretations of the input, also aligned with language.

3. **Comprehensive analysis of neural network models:** We conduct an in-depth analysis of various artificial neural network models, incorporating both vision and language inputs. Additionally, we explore different readout mechanisms and examine which models perform better in specific brain regions, while highlighting the unique advantages each provides.

The reviewers pointed out the following strengths in our paper -

1. The reviewers appreciated the thorough experimentation done with an extensive variety of DNN encoders, readout mechanisms and input stimuli (both - currently used in literature as well as novel techniques introduced by the authors) to analyse the visual cortex of the human brain.

2. The reviewers commend the introduction of a novel readout mechanism (Semantic Spatial Transformer Readout) to map encoder feature representations into brain voxel responses.

3. The reviewers appreciated the analysis describing three distinct regions in the visual cortex sensitive to varying properties of visual stimuli, which further strengthens the existing theories.

4. The reviewers also mentioned that the addition of different kinds of semantic language features is a nice contribution to the literature of LLM alignment with the visual cortex.

5. Lastly, the reviewers also appreciate the extensive literature review presented by the authors on existing work.

The reviewers agreed that the paper would be a meaningful contribution to current research modeling neural network models similar to the visual cortex if some weaknesses were addressed, and below are the major updates made based on the suggestions -

1. We agree with the reviewers that the initial version of Figure 1 (overview of our entire pipeline) was a little unorganised and confusing, and we have updated it in our latest version segregating the various components.

2. We agree with the reviewers that in the initial version of the paper, the biological motivations for the Semantic Spatial Transformer (SST) Readout were not sufficiently emphasized.  The SST readout is built on top of the Spatial-Feature Factorized Linear Readout, and spatially modulates the feature maps and spatial masks, allowing for dynamic and stimulus-dependent adjustments. Different channels in a feature map often encode distinct attributes, such as edges, textures, or shapes. By allowing channel-specific transformations, the STN can adapt to the unique geometric properties of the features represented in each channel. Unlike object classification tasks, where we can employ augmentations tailored to known invariances (e.g. rotating an image won’t change the category label) to boost predictive accuracy, the geometric invariances of voxel responses are unknown. STN enables the network to learn these invariances directly from the data, providing a crucial advantage in predicting voxel responses across diverse visual regions. Moreover, STN also allows voxel-specific spatial modulation of receptive fields (RFs), inspired by studies demonstrating the dynamic adaptability of RFs to stimulus properties - RF sizes can expand or contract based on contrast Sceniak et al. (1999) and can also shift or reshape in response to contextual or attentional influences Womelsdorf et al. (2006). Unlike fixed spatial masks used in the previous readouts, the STN employs affine transformations to capture stimulus-dependent spatial changes such as translation, scaling, rotation, and shearing. This flexibility enables more accurate modeling of neural responses that exhibit dynamic RF properties.  We have added these details in our current revision from lines 274-287 in the main text.

*Response Continued in the next comment.*

---

> ### Author Response · Authors · 2024-11-30
> **Official Comment to Area and Program Chairs (Continued)**
>
> 3. To provide a better understanding of the novel STN readout mechanism, we added an experiment focussed on better interpreting the STN readouts. We calculated the distance between the affine parameters corresponding to the spatial maps of each voxel for every image, relative to the mean affine parameters across all images (Figure 10). The L2 norm of this vector was computed for each voxel. Across all encoders, we observed that stimulus-dependent spatial shifts of the receptive field increase from lower to higher visual regions. A similar trend emerged when calculating the average spatial shifts for each channel of the feature map across images for different regions. This trend further supports the idea that higher levels of the visual cortex benefit more from learned geometric invariances and exhibit greater spatial modulation of their visual receptive fields compared to lower visual cortex regions. This modulation includes phenomena such as receptive field expansion, contraction, or shifts in response to different stimuli. We have added these in Supplementary Section A.4 and Figure 10.
>
> 4. We added an additional experiment showing the relative improvements of the STN with only affine parameters for the spatial mask vs affine parameters for only the feature maps, showing that both contribute to the improved performance of the STN readout. This is added as Supplementary Table 5.
>
> 5. We added an additional experiment to analyse if the observed differences between dense and global captioning are due to (a) the spatial subdivision of the image (Hypothesis 1) or the increased semantic detail in dense captions (Hypothesis 2). The original idea behind using dense captions was to provide spatial information in addition to semantic information in the form of captions, and subdividing the image into equal sized grids and getting captions for each grid was one of the easiest and most intuitive ways to do that. As per some of the reviewer’s suggestions, we tried generating more comprehensive single captions of the image using existing LLMs, however none of them were able to provide more information than those already present in the original MS-COCO dataset. In an attempt to densify the single captions, we thus adopted a different approach: for each image, we took the embeddings of dense captions generated for individual grid locations and averaged these embeddings to produce a single "aggregate dense caption" embedding. On comparing single caption stimuli with ‘densified’ single caption stimuli (as opposed to the dense caption approach discussed in the paper), we saw a similar trend where the higher regions of the visual cortex were better modeled by single caption stimuli. However, the transition in sensitivity from dense to single caption in the middle regions of the ventral, dorsal and lateral stream that is so clearly pronounced when using dense captions is missing when using the above ‘densified’ single captions. Further comparing ‘densified’ single captions to dense captions (as proposed in the paper), we saw that the dense captions modeled the overall visual cortex better. Hence, we do feel that adding spatial information to the dense caption is necessary for building more accurate models, be it by sub-dividing the image into grids or via any other way. This analysis has been added in Supplementary Section A.3 and Figure 9 of our paper.
>
> *Response Continued in next comment.*

---

> > ### Author Response · Authors · 2024-11-30
> > **Official Comment to Area and Program Chairs (Continued)**
> >
> > 6. We added further analysis of how the STN readouts’ performance is proportional to the channel size of the encoder feature representations, and how the trends observed across the various regions of the Visual cortex are biased on how biologically intuitive a readout is in Supplementary section A.5 and Table 6. The larger improvements for vision models stem from their feature representations having greater spatial dimensions than language models, allowing the SST to better leverage the rich spatial information available in vision models. To mitigate this, we can normalize spatial dimensions across models to ensure uniform treatment. Empirically we show below that if we reduce the spatial dimensions of the vision encoder to match those of the language encoder, that does drop the prediction performance and relative gains. The overall trend—where higher cortical areas are better modeled by language input and lower cortical areas by visual input—is consistently observed across all readouts (Fig. 4, 5, 6, 7). However, the margin distinguishing the effectiveness of the models varies slightly. Notably, as we progress from less biologically intuitive readouts to more biologically plausible ones (linear regression, Gaussian 2D, Spatial-Feature Factorized Linear Readout, and finally, the Semantic Spatial Transformer Readout), these trends become increasingly well-defined. Given that the Semantic Spatial Transformer Readout most accurately and consistently models neural responses, we rely on it to delineate regions of the visual cortex sensitive to varying kinds of stimulus information.
> >
> > 7. We highlighted the fact that ‘Task’ and ‘Response’ Optimised models as introduced in literature do not differ merely based on their training diet, but the network architecture and structural biases play an important role. The emphasis was not on directly comparing task- and response-optimized models after controlling for every factor that distinguishes them besides the objective. Instead, our goal was to select the most effective pretrained model (with optimized architecture and dataset) for task-optimized applications versus the most suitable architecture for response-optimized models that aligns with neural data constraints. This analysis is added in Supplementary Section A.6 and Table 7.
> >
> > We sincerely thank the reviewers who have engaged with our responses and acknowledge that the additional clarifications and analyses have strengthened our manuscript. We request the Area Chair to consider the highlighted strengths and contributions of our work in making a final decision regarding the paper.

---

### Author Response · Authors · 2024-12-02
**Request for Review of Updated Paper and Revisions**

We sincerely thank all the reviewers for dedicating their time and effort to provide valuable feedback on our paper. Your thoughtful suggestions have been instrumental in enhancing the quality of our work. We have carefully addressed each of your comments and made the necessary updates to our manuscript accordingly. We kindly request all reviewers to review the updated version of our paper and consider revising their scores if they find the revisions satisfactory.

---

### Meta-Review · Area_Chair_YG17 · 2024-12-21

**Metareview:**

The reviewers collectively noted that the paper suffers from subpar presentation quality, as highlighted by reviewer gWb8, with cluttered and confusing figures and unclear descriptions. Furthermore, while the proposed Semantic Spatial Transformer readout shows improvements in prediction accuracy, its biological motivation and broader relevance remain underemphasized and inadequately justified. Reviewer EMZd pointed out that many findings replicate prior work, with limited additional insights, and raised concerns about the marginal gains of the proposed readout given its complexity. Overall, the weaknesses in writing, clarity, and limited novelty outweigh the paper’s contributions.

**Additional Comments On Reviewer Discussion:**

During the rebuttal, all reviewers raised concerns about presentation quality, the biological relevance and marginal gains of the proposed Semantic Spatial Transformer (SST) readout, and the novelty of findings compared to prior work. The authors responded by revising figures (for gWb8), adding biological justifications for the SST readout (for EMZd and rgy2), and conducting additional experiments to clarify the distinctions between task- and response-optimized models (for EMZd and zmHZ). While these efforts were acknowledged, the persistent issues in clarity (gWb8), unresolved biases in the readout method (rgy2), and limited novelty (EMZd and zmHZ) ultimately led to the recommendation against acceptance which I also agree with.

---

### Decision · Program_Chairs · 2025-01-22

Reject